# Theory of oblique topological insulators

Benjamin Moy$^\phi$, Hart Goldman$^{a_\mu}$, Ramanjit Sohal$^{b_{\mu\nu}}$, and Eduardo Fradkin$^\phi$

$^\phi$*Department of Physics and Institute for Condensed Matter Theory,*
*University of Illinois at Urbana-Champaign, Urbana, IL 61801, USA*

$^{a_\mu}$*Department of Physics, Massachusetts Institute of Technology, Cambridge, MA 02139, USA*

$^{b_{\mu\nu}}$*Department of Physics, Princeton University, Princeton, NJ 08544, USA*

December 12, 2022

## Abstract

A long-standing problem in the study of topological phases of matter has been to understand the types of fractional topological insulator (FTI) phases possible in 3+1 dimensions. Unlike ordinary topological insulators of free fermions, FTI phases are characterized by fractional Θ-angles, long-range entanglement, and fractionalization. Starting from a simple family of $\mathbb{Z}_N$ lattice gauge theories due to Cardy and Rabinovici, we develop a class of FTI phases based on the physical mechanism of oblique confinement and the modern language of generalized global symmetries. We dub these phases *oblique topological insulators*. Oblique TIs arise when dyons—bound states of electric charges and monopoles—condense, leading to FTI phases characterized by topological order, emergent one-form symmetries, and gapped boundary states not realizable in 2+1-D alone. Based on the lattice gauge theory, we present continuum topological quantum field theories (TQFTs) for oblique TI phases involving fluctuating one-form and two-form gauge fields. We show explicitly that these TQFTs capture both the generalized global symmetries and topological orders seen in the lattice gauge theory. We also demonstrate that these theories exhibit a universal "generalized magnetoelectric effect" in the presence of two-form background gauge fields. Moreover, we characterize the possible boundary topological orders of oblique TIs, finding a new set of boundary states not studied previously for these kinds of TQFTs.

# 1 Introduction

There has been much interest over the past two decades in symmetry-protected topological (SPT) phases of free fermions, so much so that many simple models have been developed, and a complete classification of such states has been achieved [1–3]. The classic example of a 3D SPT phase is the 3D time-reversal invariant topological insulator (TI), which has a bulk response given by $U(1)$ gauge theory with theta angle, $\Theta = \pi$ [4], and has surfaces hosting a single free massless Dirac fermion [5]. In addition, using functional bosonization, a hydrodynamic effective field theory with a theta term for a fluctuating $U(1)$ gauge field has been developed [6, 7]. Bosonic analogues of TIs have also been constructed, which generically host interacting boundary states [8–11].

The study of 3D topological phases involving strong interactions, long-range bulk entanglement, and topological order is far less mature. Such symmetry-enriched topological (SET) phases are governed by an interplay of symmetry protection and topological order, and may be thought of as generalizations of the fractional quantum Hall effect to 3D [12, 13]. Of special interest are *fractional* topological insulators (FTIs), a class of SET phases in which a 3D bulk is described by a gauge theory with fractional $\Theta$-angle. However, attempts to construct models of FTIs have relied either on specially tailored lattice models [14–17] or parton constructions [18–22], where the charge fractionalization is put in by hand, and the mean field behavior of the partons closely parallels the physics of electrons in a non-interacting TI. It is therefore of great importance to understand whether other microscopic mechanisms and models can realize FTI physics.

Here we focus on a lattice gauge theory originally developed by Cardy and Rabinovici [23, 24] that involves a $\mathbb{Z}_N$ gauge field on a 4D Euclidean lattice, with a lattice analogue of a $\Theta$-term. As a result, test magnetic charges acquire an electric charge by the Witten effect [25], hence becoming *dyons*. In ordinary gauge theories, the condensation of monopoles corresponds to confinement of charges [26, 27], but in the presence of a $\Theta$-term, the condensation of one type of dyon leads to confinement of others. This phenomenon is referred to as *oblique confinement* [28], and it gives rise to a rich phase diagram of different oblique confining phases. In analogy with the fractional quantum Hall effect, the global phase diagram is organized by the modular group, $\mathrm{PSL}(2, \mathbb{Z})$, which here is generated by periodicity of $\Theta$ and exchange of electric and magnetic charges. A similar structure was proposed later as a phenomenological explanation of the superuniversality of quantum Hall transitions in two-dimensional electron fluids in large magnetic fields [29–31]. Although it has since been understood [32–34] that oblique confining phases of the Cardy-Rabinovici (CR) model possess

**UV Lattice gauge theory**

Emergent one-form symmetry: $G$

**Oblique confinement:**
Symmetry breaking, $G \to G_{\text{oblique}}$

**IR Oblique TI**

**Topological order:**
$G_{\text{gauge}} = G/G_{\text{oblique}}$

**Figure 1:** Emergence of the oblique TI phase from a UV lattice gauge theory. When a dyon condenses, the emergent one-form symmetry, $G$, is broken to a non-trivial subgroup, $G_{\text{oblique}} \subset G$. The resulting topological order is that of a (zero-form) gauge theory with gauge group, $G_{\text{gauge}} = G/G_{\text{oblique}}$, but now equipped with the residual global symmetry, $G_{\text{oblique}}$.

anyonic braiding of point charges with vortex lines—and thus topological order—a detailed understanding of the precise topological order, boundary states, and topological response for a generic oblique confining phase has been lacking.

In this work, we show that oblique confinement represents a natural microscopic mechanism for realizing FTI phases using the illuminating example of the CR model. Unlike previous FTI models, generic oblique confining phases of the CR model are not traditional SPT phases protected by an ordinary global symmetry. Rather, we find that they constitute FTI phases characterized by an emergent global one-form symmetry [35], under which the charged objects are dyon world lines. This symmetry is a particular instance of the *generalized global symmetries* [35] that have attracted much recent attention in high energy and condensed matter physics. We explicitly derive a low-energy effective topological quantum field theory (TQFT) for these models, in which a two-form hydrodynamic field, $b_{\mu\nu}$, corresponding to monopole flux, couples to the flux of an emergent gauge field, $a_\mu$, as

$$S_{\text{eff}} = \int \left( \frac{iK}{4\pi} b \wedge b + \frac{ip}{2\pi} b \wedge da \right) , \qquad \Theta = -2\pi \frac{p^2}{K} , \qquad (1.1)$$

where $\Theta/2\pi = -p^2/K$ is a rational number in the oblique confining phase. Note that

while a Maxwell term for $a_\mu$ is allowed by power counting, we show in this work that its presence can be neglected deep in the oblique confining phase. We refer to states described by Eq. (1.1) as *oblique topological insulators*, and they generically display both topological order and symmetry protection. A crucial feature of oblique TI phases is the existence of an emergent one-form symmetry, which is only partially broken by the bulk topological order. The presence of a remaining *unbroken* one-form symmetry in the bulk is connected to many of their exotic universal properties (see Figure 1).

Theories of the type in Eq. (1.1) have been explored in some detail in work on generalized symmetries [35–38] and as a continuum description of Walker-Wang models [15, 16, 32], and aspects of their relationship with oblique confinement [39] and FTIs [15, 40, 41] have been discussed in the past. Here we present an explicit derivation of such theories for general oblique confining phases of the CR model, and we show that the action of modular symmetry, $\mathrm{PSL}(2, \mathbb{Z})$, on these theories allows one to traverse the global phase diagram of CR models.

In the presence of a boundary, there is a gauge anomaly associated with gauge transformations of $b_{\mu\nu}$ and $a_\mu$, necessitating the introduction of new gauge fields on the boundary. As a result, oblique TIs host exotic topologically ordered boundary states that are analogues of fractional quantum Hall phases not realizable in 2D alone. In fact, we show that there are distinct possible boundary terminations preserving different global one-form symmetries of the bulk theory, resulting in a circumstance where multiple boundary topological orders are possible for the same bulk oblique TI phase. In particular, each such boundary state can be characterized by a different fractional Hall conductivity. We further show that these different boundary states can be interpreted in terms of "electromagnetic duality," or exchanging the roles of charges and monopoles.

Finally, in analogy with the quantized magnetoelectric effect of ordinary TIs, we show that the oblique TI phases described by Eq. (1.1) possess a universal response to two-form probe fields, $B_{\mu\nu}$, of the bulk one-form symmetry,

$$S_{\text{response}} = \frac{i\Theta}{8\pi^2} \int B \wedge B + \cdots . \tag{1.2}$$

Such response generalizes the ordinary magnetoelectric effect to FTIs governed by emergent one-form symmetries. Similar types of generalized responses have also been discussed in higher dimensional examples [42]. Note that while Eq. (1.2) is a universal index describing the bulk theory, it does not correspond to a unique boundary state. Indeed, two different oblique TI phases may have the same bulk topological order but different boundary Hall conductivities.

We emphasize that, unlike the types of FTI phases discussed in the past, which exhibit a

fractional magnetoelectric effect for ordinary electromagnetic (EM) fields, oblique TIs instead exhibit a fractional response to probes of an emergent one-form symmetry, as they lack a global $U(1)$ charge conservation symmetry in the bulk. Indeed, if one wishes to think of the CR model as arising at low energies from some more microscopic model of interacting particles with unit charge, then the one-form symmetry is absent at that ultraviolet (UV) scale and should be regarded as an emergent, low-energy symmetry. We view the presence of a fractional response to emergent global symmetries as one of the unique and novel features of oblique TIs, and for our purposes we will view the "microscopic" global symmetries as those of the CR model, which has an exact $\mathbb{Z}_N$ one-form symmetry.

We proceed as follows. In Section 2, we review the physics of the Witten effect and oblique confinement. In Section 3, we introduce the CR model and review its global symmetries and phase diagram. In Section 4, we describe the topological orders of the oblique confining phases of the CR model. We then explicitly show in Section 5 that the different oblique confining phases of the CR lattice model are captured by Eq. (1.1), and we discuss the properties of these theories in detail. In Section 6, we develop the notion of the higher-form magnetoelectric effect, Eq. (1.2), in oblique TI phases. We then derive possible boundary states in Section 7. Finally, in Section 8, we discuss electromagnetic duality in the lattice model and TQFT, and we describe the action of $\mathrm{PSL}(2, \mathbb{Z})$ on these models. In our appendices, we review generalized global symmetries (Appendix A), present canonical quantization of the effective TQFT (Appendix B), discuss the effective field theory of the fermionic Cardy-Rabinovici model (Appendix C), examine how the boundary states transform under electromagnetic duality (Appendix D), and determine the effective TQFT and boundary states of the 1+1-D analogue of the Cardy-Rabinovici model, which has parafermion boundary modes (Appendix E).

## 2  Oblique confinement and generalized symmetries

The physical mechanism underlying the FTI phases we develop in this work is known as oblique confinement. The concept of oblique confinement was introduced by 't Hooft in the context of $SU(N)$ gauge theory with a $\Theta$-term [28]. Here we review the basic physics of oblique confinement in continuum models. Throughout we use the modern language of generalized global symmetries [35], reviewed in Appendix A. We additionally note that there is an analogue of oblique confinement in 1+1-D [23, 24], which we review in Appendix E. There, we also determine a corresponding 1+1-D effective field theory and discuss its boundary states, which have parafermions.

Consider a 4D $U(1)$ gauge theory with a $\Theta$-term, broken to a $\mathbb{Z}_N$ subgroup by the condensation of a charge $N$ complex scalar field with conserved current, $J_\mu$.

$$S = \int d^4x \left[ -\frac{1}{4g^2} f_{\mu\nu} f^{\mu\nu} + \frac{N\theta}{32\pi^2} \varepsilon^{\mu\nu\lambda\sigma} f_{\mu\nu} f_{\lambda\sigma} + N\, J^\mu a_\mu + \dots \right], \tag{2.1}$$

where the bulk $\Theta$-angle is $\Theta = N\theta$ and $\dots$ denote additional terms in the matter action. Deep in the condensed regime, $J_\mu$ fluctuates wildly, Higgsing $a_\mu$ to $\mathbb{Z}_N$. Below, we will denote the dual field strength as $\tilde{f}^{\mu\nu} = \frac{1}{2}\varepsilon^{\mu\nu\lambda\sigma} f_{\lambda\sigma}$. This theory has the remarkable property, known as the Witten effect [25], that a magnetic monopole of charge $q_m$ inserted into the bulk acquires an electric charge[1],

$$q_e' = \frac{\theta}{2\pi}\, q_m\,. \tag{2.2}$$

A simple way of seeing this involves considering the effect on Maxwell's equations of adiabatically turning on $\theta$ from $\theta(t=0) = 0$ to $\theta(t=t_0) = \theta$ over some time, $t_0$. As a result, any charge-monopole composite, known as a *dyon*, having charges $(q_e, q_m)$ in a vacuum ($\theta = 0$) will inherit a charge,

$$(q_e', q_m) = \left( q_e + \frac{\theta}{2\pi}\, q_m, q_m \right)\,. \tag{2.3}$$

Because any monopole inserted into the system becomes a dyon, it is natural in any system with $\theta \neq 0$ to consider phases in which some dyons condense and others confine, a phenomenon known as *oblique confinement*.

Consider the effect of condensing dyons of charge $(n, m)$. Condensing this dyon means that any dyon of charge $(q_e, q_m)$ that divides $(n, m)$ is deconfined, while any charges not in units of the condensed dyon charge are confined [23, 43]. In other words, a test charge, $(q_e, q_m)$, is deconfined if it satisfies

$$q_m\, n - q_e\, m = 0\,. \tag{2.4}$$

The concept of confinement due to condensation of dyons is known as oblique confinement. This notion takes on life particularly in $\mathbb{Z}_N$ theories, where only a discrete number of charges, given by $L = \gcd(Nn, m)$, satisfy Eq. (2.4) (see Figure 2). Thus, oblique confinement results in a *different* state from the deconfined phase of ordinary $\mathbb{Z}_N$ gauge theory, with $L$ distinct

---

[1]Note that we write electric charges in units of $N$, the dynamical charge in the UV theory in Eq. (2.1). The dyon $(q_e, q_m)$ thus has electric charge $Nq_e \in \mathbb{Z}$ and magnetic charge $q_m \in \mathbb{Z}$.

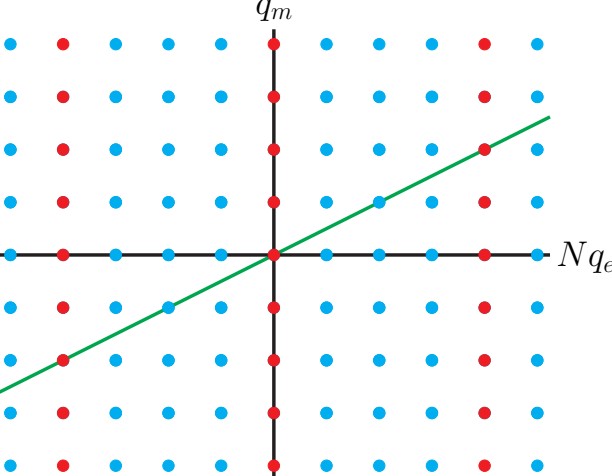

**Figure 2:** Charge lattice for the example of $\mathbb{Z}_4$ gauge theory (i.e., $N = 4$), representing possible electric charges, $Nq_e \in \mathbb{Z}$, and magnetic charges, $q_m \in \mathbb{Z}$. The red points denote dynamical charges in the UV theory, and the blue dots represent the left over "static" probes. When a dyon with charges $(q_e, q_m) = (1, 2)$ is condensed, the dyons that lie along the green line are deconfined, and all others are confined.

deconfined loop operators. Indeed, this state has a bulk topological order resembling the deconfined phase of ordinary $\mathbb{Z}_L$ gauge theory (with $\theta = 0$)[2].

Oblique confining phases have interesting global properties when viewed through the lens of generalized global symmetries [35, 36]. (See Appendix A and Ref. [45] for reviews on generalized global symmetries intended for a condensed matter audience.) The theory in Eq. (2.1) possesses a global electric $\mathbb{Z}_N$ one-form symmetry, which acts on Wilson loops, $W_\Gamma = e^{i \oint_\Gamma a}$, as

$$a_\mu \to a_\mu + \alpha_\mu/N \,, \qquad W_\Gamma \to e^{i \oint_\Gamma \alpha/N} W_\Gamma \,, \tag{2.5}$$

where $\alpha$ is a flat connection satisfying $\oint_\Gamma \alpha \in 2\pi\mathbb{Z}$ and $\Gamma$ is a closed loop. This symmetry should be viewed as *emergent*: it is explicitly broken by any gapped degrees of freedom with charge smaller than $N$, which we always allow to exist at high energies.

When dyons with charge $(n, m)$, $n, m \in \mathbb{Z}$, condense, the global properties of the gauge theory are reorganized at long wavelengths. The charge-$N$ electric charges become energetically costly: They are now confined and are projected out of the spectrum. Instead, the

---

[2]As we will discuss in Section 4.1.3, since magnetic charge can transmute statistics of electric charges [44], in certain cases, the bulk topological order can be a "twisted" $\mathbb{Z}_L$ gauge theory, which contains deconfined fermionic point excitations. This situation arises when the dyon $(n/L, m/L)$ is a fermion. Otherwise, if $(n/L, m/L)$ is bosonic, the bulk topological order is the same as the deconfined phase of ordinary $\mathbb{Z}_L$ gauge theory.

low energy charged fluctuations—the dyons—have electric charge, $Nn$, as in Figure 2. This results in an emergent $\mathbb{Z}_{Nn}$ one-form symmetry[3], which is larger than the original $\mathbb{Z}_N$. This new emergent global symmetry is broken in the IR, in the sense that there are deconfined loops that transform non-trivially under Eq. (2.5). Because the minimally charged deconfined dyon has electric charge $Nn/L$, the residual global one-form symmetry associated with the confined dyon loops is

$$G_{\text{oblique}} = \mathbb{Z}_{Nn/L} . \tag{2.6}$$

The topological order realized by the $L$ deconfined loops is that of $G_{\text{gauge}} = \mathbb{Z}_N/\mathbb{Z}_{Nn/L} = \mathbb{Z}_L$ gauge theory. The presence of the remaining one-form global symmetry, $G_{\text{oblique}}$, distinguishes an oblique confining phase from the deconfined phase of an ordinary $\mathbb{Z}_P$ gauge theory with $\theta = 0$ and $P = L$, which has essentially the same topological order. Such a phase would correspond to the choice of $N = P$, $n = 1$, and $m = 0$, meaning that $L = \gcd(P, 0) = P$, and the residual global one-form symmetry then becomes $G_{\text{oblique}} = \mathbb{Z}_1$, which is trivial. In other words, the $\mathbb{Z}_P$ one-form symmetry is broken completely.

These general considerations indicate that oblique confinement leads to both topological order and non-trivial emergent global symmetries, and one of the goals of this work is to concretely establish the universal topological and global content of these states, as encoded in an effective topological quantum field theory, and a theory of their response. Moreover, we will see that the existence of a residual global one-form symmetry, $G_{\text{oblique}}$, means that these topological orders also exhibit symmetry protection, hence furnishing a new type of FTI phase that we dub an *oblique TI*. Indeed, these properties together present a clear paradigm for oblique TIs as FTI phases equipped with an emergent global one-form symmetry. We now proceed to develop this paradigm starting from $\mathbb{Z}_N$ lattice gauge theory models at $\theta \neq 0$ originally studied by Cardy and Rabinovici.

## 3  The Cardy-Rabinovici model

### 3.1  Lattice gauge theory and Coulomb gas representation

Our focus in this work is on a class of lattice gauge theory models first studied by Cardy and Rabinovici [23, 24]. Consider a compact $U(1)$ lattice gauge theory on a 4D Euclidean

---

[3]There is also an emergent $\mathbb{Z}_m$ magnetic one-form symmetry associated with the monopoles. This symmetry has a mixed anomaly with the electric $\mathbb{Z}_{Nn}$ one-form symmetry, and we will discuss it further in Section 4.

hypercubic lattice, with gauge field $a_\mu$ living on links, $\ell$. Coupling the theory to a charge-$N$ matter current, $n_\mu \in \mathbb{Z}$, in a condensed phase spontaneously breaks $U(1) \to \mathbb{Z}_N$. The properties and phase structure of $\mathbb{Z}_N$ gauge theory (without a $\Theta$-angle) were first studied in Refs. [46–50]. In the Villain approximation, the partition function we consider has both Maxwell and $\Theta \equiv N\theta$ terms[4],

$$Z = \int [da_\mu] \sum_{\{n_\mu, s_{\mu\nu}\}} \delta(\Delta_\mu n_\mu) \, e^{-S[n_\mu, a_\mu, s_{\mu\nu}]} \,, \tag{3.1}$$

$$S = \frac{1}{4g^2} \sum_P F_{\mu\nu}^2 + \frac{iN\theta}{16\pi^2} \sum_{r,R} F_{\mu\nu}(r) \, K(r-R) \, \widetilde{F}_{\mu\nu}(R) - iN \sum_\ell n_\mu a_\mu \,. \tag{3.2}$$

Here $F_{\mu\nu} = \Delta_\mu a_\nu - \Delta_\nu a_\mu - 2\pi s_{\mu\nu}$ is the lattice field strength, valued on plaquettes of the direct lattice (denoted $P$), which has sites $r$, and $\widetilde{F}_{\mu\nu} = \frac{1}{2}\varepsilon_{\mu\nu\lambda\sigma} F_{\lambda\sigma}$ is the dual field strength, valued on plaquettes of the dual lattice, which has sites $R$. The integer variables, $s_{\mu\nu} \in \mathbb{Z}$, are defined on plaquettes and enforce compactness of the gauge theory such that we may allow $a_\mu \in \mathbb{R}$. The first term is the usual Maxwell term of lattice QED, with coupling strength, $g$. The second term is meant to be a lattice approximation to the $\Theta$-term, and thus involves a non-local product of field strengths on the direct and dual lattices. The non-locality is encoded in the short-ranged kernel, $K(r-R)$, normalized such that $\sum_x K(x) = 1$ [23]. Since we do not expect the precise form of $K(x)$ to affect any physics in the continuum limit, we assume it to simply give a delta function in the continuum.

The partition function is invariant under $U(1)$ gauge transformations,

$$a_\mu \to a_\mu + \Delta_\mu \chi. \tag{3.3}$$

We may therefore define the gauge invariant Wilson loop operator,

$$W_\Gamma = \prod_{\ell \in \Gamma} e^{i \, a(\ell)} \,, \tag{3.4}$$

which is the basic gauge invariant observable of the theory.

This lattice gauge theory contains both dynamical charges and monopoles. In Eq. (3.1), the integer-valued world line variables, $n_\mu$, represent dynamical charge-$N$ electric matter[5].

---

[4]Aspects of four-dimensional Abelian lattice gauge theories with $\Theta$-terms (in particular, the presence or absence of periodicity in the $\Theta$-angle) have been discussed recently in Ref. [51] and references therein.

[5]If the underlying microscopic matter degrees of freedom described by $n_\mu$ are fermions, then $N$ must be even (only even numbers of fermions can form a boson and condense), but if they are bosons, then $N$ can be even or odd. We will primarily focus in this work on the case of bosonic matter unless otherwise noted (all of our results can be extended to the case of fermionic matter, see Appendix C).

By summing over the integer-valued fields $n_\mu$, we see that the vector potential $a_\mu$ is Higgsed such that $\exp(ia_\mu)$ is an $N^{\text{th}}$ root of unity and hence takes values in $\mathbb{Z}_N$. As a result, the theory is invariant under a global $\mathbb{Z}_N$ electric one-form symmetry,

$$a_\mu \to a_\mu + \frac{2\pi}{N}\eta_\mu, \tag{3.5}$$

where $\eta_\mu \in \mathbb{Z}$ and $\Delta_\mu\eta_\nu - \Delta_\nu\eta_\mu = 0$. This global symmetry acts on Wilson loops as

$$W_\Gamma \to \omega\, W_\Gamma, \tag{3.6}$$

where $\omega$ is an $N^{\text{th}}$ root of unity, and hence we will say that $\omega \in \mathbb{Z}_N$. In addition, as in compact QED, the theory contains dynamical integer-valued magnetic monopoles, with world lines given by the flux of the integer-valued Kalb-Ramond fields, $s_{\mu\nu}$,

$$m_\mu = \frac{1}{2}\varepsilon_{\mu\nu\lambda\sigma}\Delta_\nu s_{\lambda\sigma}\,. \tag{3.7}$$

Therefore, $s_{\mu\nu}$ may be thought of as a configuration of world sheets of Dirac strings, which end on monopoles with world lines described by the integer variables, $m_\mu$.

To determine the phase diagram of the theory, it is useful to pass to the Coulomb gas representation by integrating out the gauge field, $a_\mu$. This leads to a new effective theory of interacting charges and vortices,

$$
\begin{aligned}
Z = \sum_{\{n_\mu, m_\mu\}} \exp\Bigg[ &-\frac{2\pi^2}{g^2}\sum_{R,R'} m_\mu(R)\,\mathcal{G}(R-R')\,m_\mu(R') \\
&-\frac{1}{2}N^2 g^2 \sum_{r,r'} \left(n_\mu(r) + \frac{\theta}{2\pi}m_\mu(r)\right)\mathcal{G}(r-r')\left(n_\mu(r') + \frac{\theta}{2\pi}m_\mu(r')\right) \\
&+iN\sum_{R,r} m_\mu(R)\,\Theta_{\mu\nu}(R-r)\,n_\nu(r)\Bigg],
\end{aligned}
\tag{3.8}
$$

where $\mathcal{G}$ is the lattice propagator for $a_\mu$ in Feynman gauge ($\Delta^2 a_\mu = 0$), and $\Theta_{\mu\nu}$ is an angle defined as [23]

$$\Theta_{\mu\nu}(R-r) = 2\pi\,\varepsilon_{\mu\nu\lambda\sigma}\,u_\lambda(u\cdot\Delta)^{-1}\Delta_\sigma^{(r)}\mathcal{G}(R-r) \tag{3.9}$$

for a unit vector $u$. The first term in Eq. (3.8) represents interactions between monopoles, and the second term describes interactions of electric charges, which notably are shifted from $n_\mu$ by the Witten effect, $n_\mu \to n_\mu + (\theta/2\pi)m_\mu$.

The final term in Eq. (3.8) is the most important and describes a statistical interaction between the electric charges and vortices depicted in Figure 3. In 3+1-D, vortices are lines in space and may be understood as Dirac strings ending on monopoles. Whenever an electric charge crosses the world sheet of a vortex, the angle $\Theta_{\mu\nu}$ changes by $2\pi$.

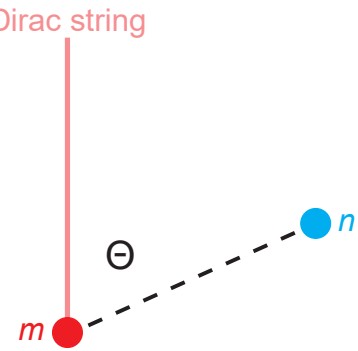

**Figure 3:** The statistical interaction in the Coulomb gas representation. If the monopole current, $m_\mu(R)$, describes a vortex string extending vertically in the $z$-direction, and $n_\mu(r)$ describes a static charge with world line in the Euclidean "time" direction, $\tau$, then $\Theta_{\mu\nu}(R-r)$ is the angle $r-R$ forms with the vortex world sheet in the $(x,y)$-plane.

## 3.2 Oblique confinement in the Cardy-Rabinovici model

Equipped with the Coulomb gas representation, one can see that the model, Eq. (3.1), displays a rich array of oblique confining phases by studying the minima of the free energy. We show in Section 4 that these phases generically exhibit topological order and are invariant under the one-form global symmetry in Eq. (2.6), making them oblique TIs.

To determine the phase diagram, Cardy and Rabinovici [23, 24] used standard free energy arguments to compare the energy cost of exciting a dyon loop to the entropy gained by the system, finding that a condensate of a dyon of charge[6] $(n, m)$ is stable if it satisfies

$$\frac{2\pi}{Ng^2}m^2 + \frac{Ng^2}{2\pi}\left(n + \frac{\theta}{2\pi}m\right)^2 < \frac{C}{N}, \tag{3.10}$$

where $C \sim 1/\mathcal{G}(0)$ is a non-universal constant. If no dyon satisfies Eq. (3.10), as occurs only for sufficiently large $N$, then nothing condenses, and the theory is in a Coulomb phase. If multiple dyons satisfy Eq. (3.10), then the bosonic dyon with the lowest free energy condenses. Although the constant, $C$, in Eq. (3.10) is not precisely determined, the exact value does not dictate the possible phases but only establishes where the phase transitions occur. Moreover, there is a 2D analogue of the Cardy-Rabinovici model, reviewed in Appendix E, for which it is possible to derive a precise bound for the energetic stability of a given phase.

An asymptotic picture for the phase diagram can be obtained by considering limits of Eq. (3.10). In the weak coupling limit, $g^2 \to 0$, the first term in Eq. (3.10) dominates,

---

[6]We adopt a convention of denoting a dyon's charges by their values $(n, m)$ *prior* to accounting for the Witten effect and with the electric charge defined in units of the condensed charge, $N$. In this notation, the true electric charge of the dyon is then $N[n + (\theta/2\pi)m]$.

preventing dyons with any magnetic charge, $m \neq 0$, from condensing. The only possibility is for electric charges $(1, 0)$ to condense, resulting in a Higgs phase, the deconfined phase of $\mathbb{Z}_N$ gauge theory at $\theta = 0$. This phase has $\mathbb{Z}_N$ topological order and is represented by a BF topological field theory at level $N$, whose action is given in Eq. (A.8).

Because of the Witten effect, the fate of the theory in the strong coupling limit, $g^2 \to \infty$, depends on the value of $\theta$. At $\theta = 0$, the $g^2 \to \infty$ limit leads to condensation of $(0, 1)$ monopoles and confinement of electric charges. On the other hand, at $\theta \neq 0$, dyon condensation becomes preferable. For example, if there exist integers $p, q$ with $\gcd(p, q) = 1$ such that $\theta/2\pi = -p/q$, then Eq. (3.10) predicts that dyons of charge $(n, m) = (p, q)$ will condense as $g^2 \to \infty$. For finite values of $g^2$ at fixed rational $\theta \neq 0$, the theory generally passes through a sequence of oblique confining phases until the limiting phase predicted at $g^2 \to \infty$ is reached. Finally, if $\theta/2\pi$ is irrational, the theory does not approach any particular asymptotic regime as $g^2 \to \infty$, and the theory passes through an infinite sequence of oblique confining phases as $g^2$ is increased [23, 24].

Note that the above argument assumes that any condensed dyons are *bosons*, since only bosons can condense [32]. However, magnetic charge transmutes the statistics of particles, meaning that some of the dyons are actually fermions [44]. Indeed, the statistics of a particular dyon depend on whether the microscopic charge-1 matter degrees of freedom are fermions or bosons (i.e. whether the gauge field $a_\mu$ is $\mathrm{spin}_c$ or not). In a theory of microscopic fermions (the $\mathrm{spin}_c$ case), the statistical phase of a $(n, m)$ dyon is $(-1)^{Nnm+Nn}$. In this case, we must require $N$ to be even so that the charge-$N$ objects that condense to give the $\mathbb{Z}_N$ gauge theory in the first place are bosons. Consequently, the charge $(n, m)$ dyons are always bosons if the microscopic charges are fermions.

On the other hand, if the microscopic charges are bosons, a $(n, m)$ dyon has statistical phase $(-1)^{Nnm}$, so charge $(n, m)$ dyons are bosons if $Nnm$ is even and fermions if $Nnm$ is odd. As a result, for fermionic dyons, what we may have expected to be a $(n, m)$-condensed phase will instead be a superconductor in which the dyons pair to give a $(2n, 2m)$-condensed phase.

## 3.3   Modular invariance and phase diagram

A more detailed understanding of the phase diagram can be obtained by exploiting the self-duality of the Coulomb gas description. Indeed, the partition function of the CR model is invariant under a set of duality transformations that generate the modular group, $\mathrm{PSL}(2, \mathbb{Z})$ [24]. Modular transformations of the CR model act on the complex coupling

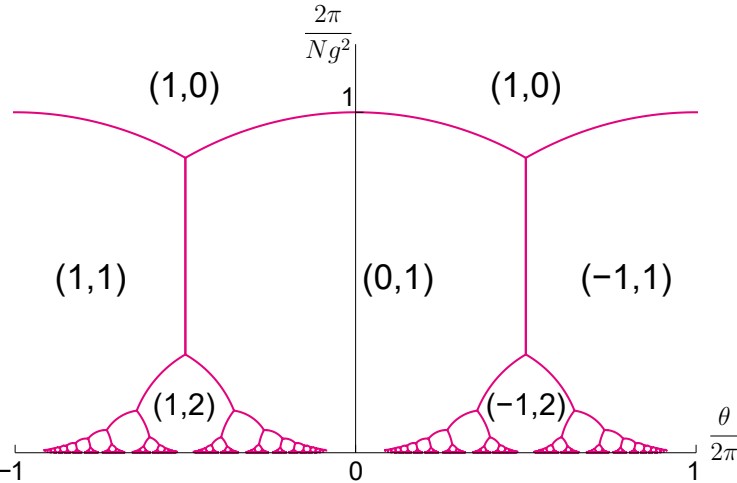

**Figure 4:** The phase diagram for the Cardy-Rabinovici model with $N/C < \sqrt{3}/2$ [23, 24]. Pink curves denote phase boundaries. The phases are labeled by $(n, m)$, indicating that a dyon of electric charge $Nn$ (without the additional polarization charge of the Witten effect) and magnetic charge $m$ condenses. For larger $N$, Coulomb phases appear, separating the gapped condensed phases.

constant,

$$\tau = \frac{\theta}{2\pi} + i\frac{2\pi}{Ng^2}. \tag{3.11}$$

In terms of $\tau$, the partition function, Eq. (3.8), in the Coulomb gas picture is

$$Z[\tau] = \sum_{\{n_\mu, m_\mu\}} \exp\left[ -\frac{2\pi i N}{\tau - \bar{\tau}} \sum_{r,r'} [n_\mu(r) + \tau\, m_\mu(r)]\, \mathcal{G}(r - r')\, [n_\mu(r') + \bar{\tau}\, m_\mu(r')] \right.$$
$$\left. + iN \sum_{R,r} m_\mu(R)\, \Theta_{\mu\nu}(R - r)\, n_\nu(r) \right], \tag{3.12}$$

where $\bar{\tau}$ is the complex conjugate of $\tau$. The partition function is manifestly invariant under the transformations[7]

$$\begin{aligned}
\mathcal{S}_{\text{CR}} : && \tau \mapsto -\frac{1}{\tau}, && (n_\mu, m_\mu) \mapsto (-m_\mu, n_\mu), \\
\mathcal{T}_{\text{CR}} : && \tau \mapsto \tau + 1, && (n_\mu, m_\mu) \mapsto (n_\mu - m_\mu, m_\mu).
\end{aligned} \tag{3.13}$$

Invariance of the partition function, Eq. (3.1), under $\mathcal{S}_{\text{CR}}$ may also be shown using the Poisson summation formula. Together, $\mathcal{S}_{\text{CR}}$ and $\mathcal{T}_{\text{CR}}$ generate the modular group $\text{PSL}(2, \mathbb{Z})$,

$$\tau \mapsto \frac{a\tau + b}{c\tau + d}, \tag{3.14}$$

---

[7]The label 'CR' for the transformations of Eq. (3.12) anticipates the fact that in Section 8 we will uncover different transformations with the same formal algebraic properties but with different physical content.

where $a, b, c, d \in \mathbb{Z}$ and $ad - bc \neq 0$.

The transformation, $\mathcal{S}_{\text{CR}}$, can be loosely understood as a kind of electromagnetic duality, as it exchanges charge and monopole world lines, and $\mathcal{T}_{\text{CR}}$ simply reflects the periodicity of $\theta$. The free energy of each oblique condensate is also invariant under these transformations, as can be seen by observing the invariance of the energy of a single dyon loop in Eq. (3.10).

Phase transitions between different oblique confining phases correspond to the so-called modular fixed point values of $\tau$, which are invariant under some modular transformation. The modular descendants of the fixed points then correspond to daughter phase transitions, leading to the rich structure of the phase diagram in Figure 4. In particular, along the $\theta = 0$ axis (for small enough $N$ such that there is no Coulomb phase), the transition between the Higgs and confinement phases occurs at the self-dual point, $\tau = i$ ($g^2 = 2\pi/N$), which is invariant under $\mathcal{S}_{\text{CR}}$. This transition then extends along the curve $|\tau| = 1$ until $\theta = \pi$, where the curve meets itself under a $\mathcal{T}_{\text{CR}}$ transformation. The rest of the phase diagram may then be constructed by repeatedly applying $\mathcal{S}_{\text{CR}}$ and $\mathcal{T}_{\text{CR}}$. Each oblique confining phase is therefore an image of the Higgs phase under some modular transformation. For any phase of condensed $(n, m)$, $n$ and $m$ are necessarily relatively prime[8] since $\gcd(n, m)$ is invariant under $\mathcal{S}_{\text{CR}}$ and $\mathcal{T}_{\text{CR}}$, and we have for the Higgs phase that $\gcd(n, m) = \gcd(1, 0) = 1$.

Throughout the rest of this work, we will refer to the duality transformations in Eq. (3.13) as the Cardy-Rabinovici (CR) modular transformations, since these transformations leave the partition function, but not necessarily correlation functions, invariant. Hence, they map distinct oblique phases into one another. As we will discuss in Section 8, a different set of modular transformations furnish genuine self-duality transformations of the CR theory.

## 4  Oblique TI phases of the Cardy-Rabinovici model

Having established the basic properties of the Cardy-Rabinovici model and its phase diagram, in this section we will now establish the topological orders and global symmetries of each of its oblique confining phases. In doing so, we will see that such states generically furnish a special class of FTI phases protected by one-form symmetries, which we call oblique TI phases.

---

[8]Some exceptions, as noted above, occur in bosonic theories. In those cases, if $Nnm$ is odd, then $(2n, 2m)$ will condense, and we will have $\gcd(2n, 2m) = 2$.

## 4.1 Oblique topological order: Lattice gauge theory

### 4.1.1 Loop operators

We begin by establishing the topological order of a phase in which a $(n, m)$ dyon condenses. The basic gauge invariant electric observables are Wilson loops,

$$W_\Gamma = \prod_{\ell \in \Gamma} e^{ia(\ell)} , \qquad (4.1)$$

where $\Gamma$ is a closed loop on the links of the Euclidean lattice. The magnetic observables are 't Hooft loops, $T_\Gamma$, which can be represented in terms of a dual gauge field, $\tilde{a}(\tilde{\ell})$, living on links of the dual lattice,

$$T_\Gamma = \prod_{\tilde{\ell} \in \Gamma} e^{i\tilde{a}(\tilde{\ell})} . \qquad (4.2)$$

The 't Hooft loops do not have a local representation in terms of fields in the "electric" theory.

Dyonic observables are built out of products of Wilson and 't Hooft loops. A dyon loop of charge $(q_e, q_m)$ can be constructed as

$$D_\Gamma(q_e, q_m) = (W_\Gamma)^{Nq_e} (T_\Gamma)^{q_m} , \qquad (4.3)$$

where we continue to express electric charge in units of $N$, the charge we condensed to obtain a $\mathbb{Z}_N$ gauge theory. This means that it is possible for $q_e$ to take fractional values that are integer multiples of $1/N$. We remark that to precisely define $D_\Gamma$, one must equip $\Gamma$ with a "framing," [52, 53] i.e. a choice of how the magnetic world line on the dual lattice tracks the electric world line on the direct lattice. However, this choice will not figure significantly into the discussion of universal properties below, and we will perform calculations ignoring the separation between the electric and magnetic world lines.

The confinement of dyons can be assessed by computing the expectation value of the dyon loop operator, $D_\Gamma(q_e, q_m)$,

$$\langle D_\Gamma(q_e, q_m) \rangle = \frac{1}{Z} \int [da_\mu] \sum_{\{n_\mu, s_{\mu\nu}\}} (W_\Gamma)^{Nq_e} (T_\Gamma)^{q_m} e^{-S[n_\mu, a_\mu, s_{\mu\nu}]} . \qquad (4.4)$$

This calculation is straightforward in the Coulomb gas representation, Eq. (3.8), where electric and magnetic variables are on equal footing. If $J_\mu$ and $\widetilde{J}_\mu$ are currents defining the

loop $\Gamma$ on the direct and dual lattices respectively, the insertion of $D_\Gamma$ into the partition function simply shifts the electric and magnetic world line variables,

$$n_\mu \to n_\mu + q_e J_\mu, \qquad m_\mu \to m_\mu + q_m \widetilde{J}_\mu. \tag{4.5}$$

In a phase in which a dyon of charge $(n, m)$ is condensed, all line operators, $D_\Gamma(q_e, q_m)$, are confined (and thus projected out of the spectrum) unless they have trivial statistical interaction with the $(n, m)$ condensate. This means that they must satisfy [23]

$$q_m\, n - m\, q_e = 0. \tag{4.6}$$

Dyon loop operators satisfying Eq. (4.6) exhibit a perimeter law. Because $q_e$ takes allowed values $q_e = p/N$, $p \in \mathbb{Z}$, and $q_m \in \mathbb{Z}$, the number of distinct deconfined loop operators is simply the greatest common divisor of $Nn$ and $m$,

$$L = \gcd(Nn, m). \tag{4.7}$$

Notice that this implies that any phase with $L = 1$ is automatically topologically trivial. Explicitly, these line operators are parameterized by a single integer, $k$,

$$\text{deconfined dyons:} \qquad D_\Gamma\left(q_e = \frac{nk}{L}, q_m = \frac{mk}{L}\right), \qquad 0 \le k < L. \tag{4.8}$$

These operators describe the physical loop excitations in the oblique confining phase where the $(n, m)$ dyon is condensed.

### 4.1.2 Surface operators

In three spatial dimensions, topological order is determined by the braiding of world lines of point particles with world sheets of flux tubes. Indeed, in addition to the Wilson and 't Hooft loops, we may also construct gauge invariant surface operators representing the world sheets of flux tubes,

$$U_\Sigma(\Phi_e) = \prod_{P \in \Sigma} e^{i\Phi_e F(P)}, \qquad \widetilde{U}_\Sigma(\Phi_m) = \prod_{\widetilde{P} \in \Sigma} e^{i\Phi_m \widetilde{F}(\widetilde{P})}, \tag{4.9}$$

where $F(P)$ and $\widetilde{F}(\widetilde{P})$ are respectively the lattice field strength and dual field strength, which live on plaquettes of the direct lattice and dual lattice. The parameters $\Phi_e$ and $\Phi_m$ set the electric and magnetic fluxes, and $\Sigma$ is a closed 2-surface.

We first consider the electric flux sheet operator. To compute its expectation value, we insert a background electric flux tube into the action, with world sheet described by a background world sheet field, $\Sigma_{\mu\nu} \in \mathbb{Z}$,

$$\frac{i\Phi_e}{2} \sum_r \Sigma_{\mu\nu}(r) \, F_{\mu\nu}(r) = \frac{i\Phi_e}{2} \sum_r \Sigma_{\mu\nu}(r) \left[\Delta_\mu a_\nu(r) - \Delta_\nu a_\mu(r) - 2\pi s_{\mu\nu}(r)\right]. \tag{4.10}$$

Because the surface is closed, we require $\Delta_\mu \Sigma_{\mu\nu} = 0$, so Eq. (4.10) reduces to

$$\frac{i\Phi_e}{2} \sum_r \Sigma_{\mu\nu}(r) \, F_{\mu\nu}(r) = -\frac{i\Phi_e}{2} \, 2\pi \sum_r \Sigma_{\mu\nu}(r) \, s_{\mu\nu}(r). \tag{4.11}$$

Similarly, we may represent magnetic flux sheet operators in terms of the dual gauge field, $\tilde{a}_\mu$, as

$$\frac{i\Phi_m}{2} \sum_R \Sigma_{\mu\nu}(R) \left[\Delta_\mu \tilde{a}_\nu(R) - \Delta_\nu \tilde{a}_\mu(R) + 2\pi \, t_{\mu\nu}(R)\right] = \frac{i\Phi_m}{2} \, 2\pi \sum_R \Sigma_{\mu\nu}(R) \, t_{\mu\nu}(R), \tag{4.12}$$

where $\tilde{a}_\mu$ is the dual of $a_\mu$, and $t_{\mu\nu}$ is defined so that $N n_\mu = \frac{1}{2}\varepsilon_{\mu\nu\lambda\sigma}\Delta_\nu t_{\lambda\sigma}$.

The insertion of a world sheet of a dyonic flux tube $(\Phi_e, \Phi_m)$ has action determined by its braiding with the dynamical charges and monopoles. To see this, we again work in the Coulomb gas representation, writing

$$s_{\mu\nu}(r) = \sum_{r'} \varepsilon_{\mu\nu\lambda\sigma}\Delta_\lambda \mathcal{G}(r - r') \, m_\sigma(r'), \tag{4.13}$$

$$t_{\mu\nu}(r) = \sum_{r'} \varepsilon_{\mu\nu\lambda\sigma}\Delta_\lambda \mathcal{G}(r - r') \, N \, n_\sigma(r'), \tag{4.14}$$

where $\mathcal{G}(r - r')$ is again the gauge fixed lattice propagator (although the expressions in Eqs. (4.10) and (4.12) are both gauge invariant), and we have dropped any distinction between the direct and dual lattices for notational brevity. A flux tube $\Sigma_{\mu\nu}$ with electric flux $\Phi_e$ and magnetic flux $\Phi_m$ therefore has action,

$$2\pi i \, \varphi\Big[\Sigma, \{Nn, m\}\Big] = 2\pi i \sum_{r,r'} \frac{1}{2} \Sigma_{\mu\nu}(r) \, \varepsilon_{\mu\nu\lambda\sigma}\Delta_\lambda \mathcal{G}(r - r')\Big\{\Phi_m[Nn_\sigma(r')] - \Phi_e[m_\sigma(r')]\Big\}. \tag{4.15}$$

The linking number, $\varphi[\Sigma, J]$, is an integer-valued topological invariant counting the linking of a configuration of closed world sheets, $\Sigma_{\mu\nu}$, with a configuration of current loops, $J_\mu$.

We may now construct the physical surface operators in the $(n, m)$-condensed phase. In this phase, the proliferating dyon loops can be described in terms of electric and magnetic world lines that track one another,

$$n_\mu = n j_\mu, \qquad m_\mu = m j_\mu, \tag{4.16}$$

where $j_\mu \in \mathbb{Z}$ is an integer-valued world line variable. Note that we continue to neglect the distinction between the direct and dual lattices, and we assume a framing of dyon loops such that electric and magnetic world-lines never cross. The action for a world sheet, $\Sigma_{\mu\nu}$, is dominated by these dyon configurations,

$$S_\Sigma = 2\pi i \left( N n\, \Phi_m - \Phi_e\, m \right) \varphi[\Sigma, j] \,. \tag{4.17}$$

Because $j_\mu$ is a dynamical variable and $\varphi$ is an integer, the expectation value of a $(\Phi_e, \Phi_m)$ surface operator will vanish (i.e. be a sum of wildly oscillating phases) unless

$$\left( N n\, \Phi_m - \Phi_e\, m \right) \in \mathbb{Z} \,. \tag{4.18}$$

In other words, physical flux tubes must braid trivially with the dyon condensate.

### 4.1.3 Topological order: Braiding of particles with flux tubes

We are now equipped to determine the topological data of oblique TI phases. We begin by computing the braiding statistics of physical flux tubes, described by a background world sheet, $\Sigma_{\mu\nu}$, with the deconfined dyon particles, described by a background world line, $J_\mu$, with charges $(q_e, q_m) = (kn/L, km/L)$. Their statistical phase, Eq. (4.15), is

$$\vartheta[\Sigma, J] = 2\pi i \left( N \frac{kn}{L} \Phi_m - \Phi_e \frac{km}{L} \right) \varphi[\Sigma, J] \,. \tag{4.19}$$

This term is the only place where the surface operators appear in the partition function, so braiding with the deconfined dyons is the only way to detect a flux tube in the low energy limit. Hence, flux tubes that have identical braiding with all the deconfined dyons are indistinguishable,

$$(\Phi_e, \Phi_m) \sim (\Phi_e + N n\, \alpha, \Phi_m + m\, \alpha), \tag{4.20}$$

where $\alpha \in \mathbb{R}$.

To count the number of inequivalent flux tubes, we can set $\Phi_e = 0$ because Eq. (4.20) implies that any flux tube can be expressed as either purely electric or purely magnetic via a suitable choice of $\alpha$. The purely magnetic flux tubes that satisfy Eq. (4.18) have $\Phi_m = M/Nn$, where $M \in \mathbb{Z}$ and $M \sim M + Nn$. However, since $\Phi_e$ is defined mod 1, we also have $M \sim M + m$ by Eq. (4.20). Therefore, the $(n, m)$-condensed phase has $L = \gcd(Nn, m)$ inequivalent flux tubes indexed by $M$.

A state with $L$ distinct line operators and $L$ distinct surface operators resembles a $\mathbb{Z}_L$ gauge theory. Indeed, from Eq. (4.19), we see that the statistical phase between the $L$ flux tubes and the $L$ deconfined dyons is

$$\left\langle D_\Gamma\left(q_e = \frac{nk}{L}, q_m = \frac{mk}{L}\right) \widetilde{U}_\Sigma\left(\Phi_m = \frac{M}{Nn}\right)\right\rangle = \exp\left(2\pi i\,\frac{kM}{L}\,\varphi[\Sigma, J]\right), \qquad (4.21)$$

where $0 \leq k < L$ and $0 \leq M < L$. Hence, so long as the $(n/L, m/L)$-dyon is a boson, the topological order is identical to $\mathbb{Z}_L$ gauge theory: the ground state degeneracy[9] and excitation spectrum are the same. When the $(n/L, m/L)$-dyon is a fermion (i.e., when $Nn/L$ and $m/L$ are both odd), the resulting paired state can have fermionic excitations, since the $(kn/L, km/L)$-dyon has statistical phase $(-1)^{Nnmk^2/L^2} = (-1)^k$. Such a theory generalizes the topological order of the "fermionic toric code" constructed in Refs. [16, 32].

## 4.2 Generalized global symmetries of oblique TIs

As discussed in Section 2, the major difference between oblique TI states of the Cardy-Rabinovici model and the deconfined phases of $\mathbb{Z}_L$ gauge theories lies in the global symmetries of the two sets of states. Because the oblique TI ground states arise from confinement of dyons, they possess residual one-form global symmetries *not* present in an ordinary $\mathbb{Z}_L$ gauge theory. As a result, one may think of these states as "one-form symmetry enriched $\mathbb{Z}_L$ gauge theories." We now turn to quantify the global one-form symmetries of oblique TI phases in the context of the CR model. We remark that topological phases enriched by generalized symmetries have received some attention recently [15, 38, 54–58], but the range of possibilities for such phases in 3+1-D has not been explored extensively. Oblique confinement provides a transparent recipe for realizing these types of states.

We first focus on the electric one-form symmetry, which acts on the Wilson loop operators, $W_\Gamma$. When a dyon with charges $(n, m)$ condenses, the theory has an emergent Gauss' Law,

$$\frac{1}{2\pi}\,\Delta_i\,E_i = 0 \bmod Nn\,, \qquad (4.22)$$

where $E_i = F_{i0}$. Consequently, there is an emergent $\mathbb{Z}_{Nn}$ one-form symmetry which preserves the condensed dyon loop operator, $D_\Gamma(n, m)$, defined in Eq. (4.3). However, we have shown above that this state has $L = \gcd(Nn, m)$ deconfined dyon loop operators, Eq. (4.8). This means that the $\mathbb{Z}_{Nn}$ one-form symmetry is broken to a subgroup in the oblique TI phase. The

---

[9]See Appendix B for an explicit calculation of the ground state degeneracy on a torus using the effective field theory we establish in Section 5.1.

remaining subgroup that preserves both the condensate and the deconfined loop operators can be read off from Eq. (4.8),

$$G_{\text{oblique}} = \mathbb{Z}_{Nn/L} \, . \tag{4.23}$$

We view this as the global symmetry characterizing the oblique TI phase, and it arises due to the fact that the number of deconfined line operators is *smaller* than the electric charge of the condensed dyon, meaning that the theory contains a residual unbroken one-form symmetry that acts trivially on all of the deconfined loops.

It is natural to consider the response of the theory to probing $G_{\text{oblique}}$, i.e. gauging the symmetry. In the Cardy-Rabinovici model, this amounts to introducing a two-form probe field, $\mathcal{B}_{\mu\nu}/N$, which couples minimally in the lattice action, i.e. we replace

$$F_{\mu\nu} \to F_{\mu\nu}[\mathcal{B}] = \Delta_\mu a_\nu - \Delta_\nu a_\mu - 2\pi s_{\mu\nu} - 2\pi \frac{\mathcal{B}_{\mu\nu}}{N} \, , \tag{4.24}$$

where $\mathcal{B}_{\mu\nu} \in \mathbb{Z}$ is a flat, integer-valued two-form field. It may be understood physically as a fractional magnetic flux that has been inserted into the theory (integer fluxes can be absorbed into the sum over $s_{\mu\nu}$). In Section 6, we will see that on condensing dyons and entering the oblique TI phase, the field, $\mathcal{B}_{\mu\nu}$, probes the residual one-form symmetry, Eq. (4.23), and leads to a universal response that is a generalization of the magnetoelectric effect in ordinary 3D TIs.

One may also consider a magnetic one-form symmetry acting on 't Hooft loops. Following the arguments leading to the electric one-form symmetry in Eq. (4.23), one finds a preserved magnetic one-form symmetry that leaves the deconfined dyon loops, Eq. (4.8), invariant,

$$\widetilde{G}_{\text{oblique}} = \mathbb{Z}_{m/L} \, . \tag{4.25}$$

It is then tempting to think that the ultimate global symmetry of the oblique TI bulk state is $G_{\text{oblique}} \times \widetilde{G}_{\text{oblique}}$. However, these electric and magnetic one-form symmetries have a mixed 't Hooft anomaly, which we will see explicitly in Section 6. Consequently, if we wish to consider response to probes (such as $\mathcal{B}_{\mu\nu}$) of either the electric or magnetic one-form symmetry, the other symmetry is explicitly broken. While there is no obvious reason physically to preference the electric one-form symmetry in this way, we will nevertheless always choose to work with probes of $G_{\text{oblique}}$ and consider $\widetilde{G}_{\text{oblique}}$ to be broken by the anomaly.

# 5 Effective field theory of oblique TIs

## 5.1 Derivation from the lattice model

Having established that the oblique confining phases of the CR model exhibit the topological order of $\mathbb{Z}_L$, $L = \gcd(Nn, m)$, gauge theory[10] and retain the emergent one-form symmetry in Eq. (4.23), we now seek an effective topological quantum field theory (TQFT) that captures these properties. In this section, we demonstrate that this effective TQFT can be constructed explicitly starting from the CR lattice model, Eq. (3.1). Note that while our focus will be on the CR model where the microscopic charges are bosons (i.e. the gauge fields are ordinary $U(1)$ gauge fields), we discuss the effective field theory for the fermionic CR model (with $\text{spin}_c$ gauge fields) in Appendix C.

We begin with the lattice CR action, Eq. (3.1), in the strong coupling limit, $g^2 \to \infty$,

$$S = -iN \sum_\ell n_\mu a_\mu + \sum_r \left( \frac{iN\theta}{8\pi^2} \, \varepsilon_{\mu\nu\lambda\sigma} \, \Delta_\mu a_\nu \, \Delta_\lambda a_\sigma - \frac{iN\theta}{2\pi} \, m_\mu a_\mu + \frac{iN\theta}{8} \, \varepsilon_{\mu\nu\lambda\sigma} \, s_{\mu\nu} s_{\lambda\sigma} \right) , \quad (5.1)$$

where we recall the definition, $m_\mu = \varepsilon_{\mu\nu\lambda\sigma} \Delta_\nu s_{\lambda\sigma}/2$. Here we have expanded out the $\Theta$-term and taken the kernel, $K(r - R)$, to be a delta function. We also suppressed the distinction between the direct and dual lattices, which will not play an important role in the arguments below. The strong coupling limit suppresses the Maxwell term, whose only role is to control the energetics determining which phase is realized at a given value of $\theta$. Indeed, the Maxwell term suppresses large dyon loops, but in a given oblique confining phase, the loops of the condensing dyon proliferate, and the Maxwell term can be safely ignored. Moreover, in Section 5.3, we will match the topological content presented in Section 4.1 to that of the TQFT developed in this section, thus providing an additional argument that does not rely on the strong coupling limit.

From the free energy arguments in Section 3.2, we recall that in the strong coupling limit, $g^2 \to \infty$, the phase where dyons with charge, $(n, m)$, $\gcd(n, m) = 1$, condense is stable at $\theta = -2\pi \, n/m$. Indeed, plugging in this value of $\theta$ and integrating out the non-compact gauge field, $a_\mu$, one finds the local constraint,

$$n \, m_\mu = m \, n_\mu. \quad (5.2)$$

Because $\gcd(n, m) = 1$, the solution to this constraint is none other than Eq. (4.16),

$$m_\mu = m \, j_\mu , \qquad n_\mu = n \, j_\mu , \quad (5.3)$$

---

[10]As discussed in Section 4.1.3, this $\mathbb{Z}_L$ gauge theory can be "twisted", containing deconfined fermions.

where $j_\mu$ is an integer-valued current[11]. Physically, the meaning of this constraint is to bind together electric charges and monopoles such that $j_\mu$ is the world line of the $(n, m)$ dyon. These dyons proliferate in the IR because there are no terms present to suppress large dyon loops.

After integrating out $a_\mu$, one arrives at an effective lattice gauge theory consisting of the two-form field, $s_{\mu\nu}$, alone, along with the constraint in Eq. (5.3),

$$Z_{\text{oblique}}^{(n,m)} = \sum_{\{s_{\mu\nu}, j_\mu\}} \delta\Big(m_\mu[s_{\nu\lambda}] - m j_\mu\Big) \exp\left((2\pi)^2 \sum_r \frac{i}{16\pi} \frac{Nn}{m} \varepsilon_{\mu\nu\lambda\sigma}\, s_{\mu\nu} s_{\lambda\sigma}\right), \tag{5.4}$$

where $\delta(x - y)$ is a Kronecker delta function defined on integer-valued lattice fields. With this form for the partition function, we may "integrate in" the constraint, $m_\mu[s] = m j_\mu$, by introducing an integer-valued field, $\widetilde{\alpha}_\mu(r) \in \mathbb{Z}$,

$$S = \sum_r \left(\frac{2\pi i}{m} m_\mu[s]\, \widetilde{\alpha}_\mu - (2\pi)^2 \frac{i}{16\pi} \frac{Nn}{m} \varepsilon_{\mu\nu\lambda\sigma}\, s_{\mu\nu} s_{\lambda\sigma}\right) \tag{5.5}$$

$$= \sum_r \left(-\frac{im}{2\pi} \frac{1}{2} \varepsilon_{\mu\nu\lambda\sigma} \frac{2\pi s_{\mu\nu}}{m} \frac{2\pi \Delta_\lambda \widetilde{\alpha}_\sigma}{m} - \frac{iNnm}{4\pi} \frac{1}{4} \varepsilon_{\mu\nu\lambda\sigma} \frac{2\pi s_{\mu\nu}}{m} \frac{2\pi s_{\lambda\sigma}}{m}\right), \tag{5.6}$$

where we have integrated by parts in the second line. Note that the notational choice of a tilde for $\widetilde{\alpha}_\mu$ is meant to emphasize that this field represents a "magnetic" gauge field, since it couples directly to the monopole current, i.e. its corresponding loop operators are 't Hooft loops. Indeed, the theory now has an emergent zero-form gauge symmetry,

$$\widetilde{\alpha}_\mu \to \widetilde{\alpha}_\mu + \Delta_\mu \chi + m \mathcal{N}_\mu, \tag{5.7}$$

where $\chi, \mathcal{N}_\mu \in \mathbb{Z}$. There is also an emergent one-form gauge symmetry acting as

$$s_{\mu\nu} \to s_{\mu\nu} - (\Delta_\mu \eta_\nu - \Delta_\nu \eta_\mu) + m \mathcal{M}_{\mu\nu}, \qquad \widetilde{\alpha}_\mu \to \widetilde{\alpha}_\mu + Nn \eta_\mu, \tag{5.8}$$

where $\eta_\mu \in \mathbb{Z}$ is a connection on links and $\mathcal{M}_{\mu\nu} \in \mathbb{Z}$ lives on plaquettes. At long wavelengths, the theory in Eq. (5.6) is topological, in the sense that it does not explicitly depend on any spacetime metric.

---

[11]For the bosonic theory, when $Nnm$ is odd, the $(2n, 2m)$ dyon is the bosonic dyon with the lowest free energy. In this case, then, the constraint Eq. (5.2) is satisfied by taking $m_\mu = 2m j_\mu$ and $n_\mu = 2n j_\mu$ so that the condensing dyon is a boson. For this case only, in what follows, $(n, m)$ should be replaced by $(2n, 2m)$.

## 5.2 Continuum TQFT

### 5.2.1 Magnetic variables

The corresponding continuum TQFT can be constructed in terms of one-form and two-form $U(1)$ gauge fields, $\tilde{a}_\mu$ and $\tilde{b}_{\mu\nu}$, which are related to the lattice fields by the correspondence,

$$2\pi\frac{\widetilde{\alpha}_\mu}{m} \to \tilde{a}_\mu\,, \qquad 2\pi\frac{s_{\mu\nu}}{m} \to \tilde{b}_{\mu\nu}\,, \tag{5.9}$$

where the coefficients are fixed by invariance under the emergent gauge symmetries in Eqs. (5.7) – (5.8), in that we require the zero (one) form gauge transformations to act on the continuum gauge field $\tilde{a}_\mu$ ($\tilde{b}_{\mu\nu}$) with unit charge. Physically, as mentioned above, one should interpret $\tilde{a}_\mu$ as the "magnetic" vector potential, the fluxes of which are sourced by the world lines of monopoles. Similarly, the world sheet field, $\tilde{b}_{\mu\nu}$, should be viewed as the world sheet variable of electric flux tubes, as its flux, $m\,\varepsilon^{\mu\nu\lambda\sigma}\partial_\nu\tilde{b}_{\lambda\sigma}/2\pi$, is the monopole current. Hence, despite starting with the "electric" formulation of the Cardy-Rabinovici model, we have in fact arrived at an effective theory in terms of magnetic variables.

In terms of these fields, the continuum TQFT may be written as

$$Z_{\text{oblique}} = \int \mathcal{D}\tilde{a}\,\mathcal{D}\tilde{b}\,e^{-\widetilde{S}[\tilde{a},\,\tilde{b}]}, \tag{5.10}$$

$$\widetilde{S} = -\frac{im}{2\pi}\int \tilde{b}\wedge d\tilde{a} - \frac{iNnm}{4\pi}\int \tilde{b}\wedge\tilde{b}. \tag{5.11}$$

We will see below that the TQFT that we have derived above encodes all of the topological data of oblique TI phases discussed in Section 4 and allows a direct way to develop a theory of their response. We remark that TQFTs of this type were first introduced in [39, 59] and were further developed in [35, 60]. They have been proposed previously as effective field theories for oblique confining phases [39], and in special cases have been related to phases the CR model [32]. However, ours is the first microscopic derivation of these models from a UV lattice gauge theory, and it is valid for any of the oblique confining phases of the CR model.

The gauge symmetries of the action, Eq. (5.11), are analogues of Eqs. (5.7) – (5.8). First, the theory is invariant under a $U(1)$ zero-form gauge transformation, $\tilde{a} \to \tilde{a} + d\xi$, where $\xi$ is a $2\pi$-periodic scalar field. In addition, there is a one-form gauge symmetry,

$$\tilde{b} \to \tilde{b} - d\lambda, \qquad \tilde{a} \to \tilde{a} + Nn\,\lambda, \tag{5.12}$$

where $\lambda$ is a $U(1)$ connection. We can see that such transformations leave the partition

function invariant by considering the variation of the action,

$$\delta_\lambda \widetilde{S} = 2\pi i m \int \frac{d\lambda}{2\pi} \wedge \frac{d\tilde{a}}{2\pi} + \pi i N n m \int \frac{d\lambda}{2\pi} \wedge \frac{d\lambda}{2\pi}. \tag{5.13}$$

On a generic closed four-manifold, the first term is an integer multiple of $2\pi i$ since $m \in \mathbb{Z}$, but the second term is an integer multiple of $2\pi i$ only if $Nnm$ is an *even* integer. Since a $(n, m)$ dyon in the bosonic CR model has a statistical phase of $(-1)^{Nnm}$, this requirement is simply the statement that the condensing dyon must be a boson. In the fermionic CR model, which involves a spin$_c$ connection, this requirement is instead $Nnm + Nn \in 2\mathbb{Z}$.

In addition, the theory in Eq. (5.11) exhibits invariance under a $\mathbb{Z}_m$ global one-form symmetry, under which

$$\tilde{a} \to \tilde{a} + \frac{1}{m} \Upsilon, \tag{5.14}$$

where $\Upsilon$ is a flat connection. Such transformations shift the action by an integer multiple of $2\pi i$ since the cycles of $\Upsilon$ are quantized, thus leaving the partition function invariant. We already encountered this global symmetry in Section 4.2 in the context of the lattice model: it is the emergent magnetic one-form symmetry that appears on condensing the $(n, m)$ dyons. In Section 6, we will see that this symmetry is broken anomalously on the introduction of two-form probes for the electric one-form symmetry, which is not manifest in this description.

### 5.2.2  Electric variables

In discussing oblique TI phases, it will be more convenient to work with the dual of the theory in Eq. (5.11), which is expressed in terms of "electric" variables. To obtain this theory, we invoke the standard electromagnetic duality procedure, in which we introduce an auxiliary two-form field, $\tilde{f}$, such that $\tilde{f} = d\tilde{a}$ is implemented as a constraint via a two-form Lagrange multiplier field, $f$, coupling as $f \wedge (\tilde{f} - d\tilde{a})$. Integrating out $\tilde{f}$ and $\tilde{a}$, one finds the dual theory [35, 60],

$$\widetilde{S} \longleftrightarrow S_{\text{eff}} = -\frac{iNn}{4\pi m} \int da \wedge da, \tag{5.15}$$

where $a_\mu$ is a $U(1)$ gauge field. This appears to be an ordinary $U(1)$ gauge theory with $\Theta$-angle, $\Theta = -2\pi Nn/m$. However, there is a subtlety here: the gauge field, $a_\mu$, introduced in the duality transformation also transforms under the gauged one-form symmetry, Eq. (5.12), as

$$a \to a - m\lambda, \tag{5.16}$$

where $\lambda$ is again a $U(1)$ connection. The fact that this symmetry is *gauged* implies that the naïve Hilbert space of the theory is in fact redundant. We can make this redundancy manifest by introducing a fluctuating $U(1)$ two-form gauge field, $b_{\mu\nu}$, via a Hubbard-Stratonovich transformation, leading to the final expression,

$$S = \frac{iNn}{2\pi} \int b \wedge da + \frac{iNnm}{4\pi} \int b \wedge b \,. \tag{5.17}$$

This theory is invariant under the gauge transformation, Eq. (5.16), along with

$$b \to b + d\lambda \,. \tag{5.18}$$

In analogy with the discussion of the magnetic theory above, here $a_\mu$ is an "electric" gauge field coupling to charge currents, and $b_{\mu\nu}$ describes world sheets of flux tubes. Note that in arriving at this duality, we have required $n, m \neq 0$. A lattice version of this duality, starting from the partition function in Eq. (5.4) coupled to the two-form probe introduced in Section 4.2, is derived in Section 6.

Similar to its magnetic dual, the electric theory in Eq. (5.17) explicitly displays a global $\mathbb{Z}_{Nn}$ one-form symmetry, under which

$$a \to a + \frac{1}{Nn} \Upsilon \,. \tag{5.19}$$

Here again $\Upsilon$ is a flat connection. This symmetry is the continuum realization of the electric one-form symmetry discussed at length in Section 4.2 in the context of the lattice model. Note that, as discussed in that section, the presence of deconfined loop operators will ultimately break this symmetry down to $G_{\text{oblique}} = \mathbb{Z}_{Nn/L}$, where, as before, $L = \gcd(Nn, m)$. In Section 6, we will will couple this symmetry to background fields, which we will see leads to a mixed 't Hooft anomaly with the $\mathbb{Z}_m$ magnetic one-form symmetry along with a higher-form generalization of the magnetoelectric effect.

To summarize, electromagnetic duality is the statement,

$$\frac{iNn}{2\pi} b \wedge da + \frac{iNnm}{4\pi} b \wedge b \quad \longleftrightarrow \quad -\frac{im}{2\pi} \tilde{b} \wedge d\tilde{a} - \frac{iNnm}{4\pi} \tilde{b} \wedge \tilde{b} \,. \tag{5.20}$$

Comparing the two sides of Eq. (5.20), we observe that electromagnetic duality acts by exchanging

$$\text{EM duality:} \quad Nn \to m \,, \quad m \to -Nn \,. \tag{5.21}$$

This transformation is notably *not* equivalent to the duality transformation introduced by Cardy and Rabinovici, which we discussed in Section 3.3 and which maps $n \to m$ and

$m \to -n$. Although that transformation leaves the free energy invariant, it does not preserve the topological order and thus maps one oblique TI phase of the CR model to another. On the other hand, Eq. (5.21) is a genuine duality in the sense that it provides two equivalent descriptions of the same oblique TI phase. We will discuss the difference between electromagnetic duality and the CR duality transformation in more detail in Section 8.

## 5.3 Oblique topological order: TQFT

We are now prepared to discuss the topological order associated with the dual TQFTs in Eq. (5.20). Since these theories have been discussed at length in Refs. [16, 32, 35, 38–41, 60, 61], our primary contribution will be to demonstrate that their observables map precisely onto the operators discussed in the context of the CR lattice model in Section 4.1. This constitutes a proof that the topological orders of the oblique confining phases of the CR model are described by this class of TQFTs. Importantly, this type of argument extends beyond the strong coupling limit used in Section 5.1.

We begin by constructing the gauge invariant loop operators in terms of electric variables in Eq. (5.17). A Wilson loop of the field, $a_\mu$, by itself is not gauge invariant under the one-form gauge symmetry in Eq. (5.16), but it can be made gauge invariant by attaching an operator supported on an open surface, $\Sigma$, bounded by the loop. We then consider the gauge invariant operator,

$$\mathcal{O}_\Sigma = \exp\left(i \oint_{\partial\Sigma} a + im \int_\Sigma b\right). \tag{5.22}$$

Because this operator depends on the topology of the surface, $\Sigma$, it generally has trivial correlation functions. A non-trivial loop operator can be obtained by raising $\mathcal{O}_\Sigma$ to the power $Nn/L$, where $L = \gcd(Nn, m)$, such that the surface part of the operator becomes invisible. This can be seen by noting that the equation of motion for $a$ constrains $b$ to be a $\mathbb{Z}_{Nn}$ gauge field. Such operators are referred to in the literature as "genuine" loop operators [60], and they furnish the topological content of the theory. They are generated by

$$\mathcal{D}_\Gamma = (\mathcal{O}_\Sigma)^{Nn/L}, \tag{5.23}$$

where $\Gamma = \partial\Sigma$. Since $(\mathcal{D}_\Gamma)^L = 1$, we see that there are $L$ inequivalent genuine loop operators.

The operators generated by powers of $\mathcal{D}_\Gamma$ represent the world lines of the $L$ deconfined dyons, and they are equivalent to the operators constructed in the lattice model, Eq. (4.8) (hence the similar notation). This is simply the statement that electric flux tubes are the

vortices of the magnetic gauge field, $b = d\tilde{a}/Nn$. Thus, the deconfined dyons are products of Wilson and 't Hooft loops,

$$\text{deconfined dyons:} \qquad (\mathcal{D}_\Gamma)^k = \exp\left( i\frac{Nnk}{L} \oint_\Gamma a + i\frac{mk}{L} \oint_\Gamma \tilde{a} \right), \qquad (5.24)$$

where $0 \le k < L$. These are precisely the continuum versions of the lattice gauge theory operators, $D_\Gamma \left( q_e = \frac{nk}{L}, q_m = \frac{mk}{L} \right)$, constructed in Eq. (4.8), and they have the same braiding properties.

In addition to the $L$ line operators, there are also $L$ inequivalent gauge invariant surface operators, which can be expressed in terms of $b$,

$$\mathcal{U}_\Sigma = \exp\left( i \oint_\Sigma b \right), \qquad (5.25)$$

where $\Sigma$ is a closed surface. Like the $\widetilde{U}_\Sigma$ operators in the lattice model, these operators represent world sheets of magnetic flux tubes in the oblique confining phase. As was also the case in the discussion of the surface operators in Section 4.1.3, there is an equivalence between electric and magnetic surface operators in the TQFT, and the operator $\mathcal{U}_\Sigma$ generates the complete set of surface operators.

Given the dyon loop operators, $\mathcal{D}_\Gamma$, and the surface operators, $\mathcal{U}_\Sigma$, of the TQFT, we can determine their self and mutual statistics [41, 61]. The line operator, $\mathcal{D}_\Gamma$, represents the world line of a particle with statistical phase $(-1)^{Nnm/L^2}$, and $\mathcal{U}_\Sigma$ has trivial self statistics. The mutual statistics are captured by correlation functions of $\mathcal{D}_\Gamma$ and $\mathcal{U}_\Sigma$,

$$\left\langle (\mathcal{D}_\Gamma)^k (\mathcal{U}_\Sigma)^M \right\rangle = \exp\left( 2\pi i \frac{kM}{L} \varphi[\Sigma, \Gamma] \right), \qquad (5.26)$$

where $\varphi[\Gamma, \Sigma]$ is the linking number of $\Gamma$ and $\Sigma$. This result matches that obtained using the lattice model, Eq. (4.21), and we thus conclude that the topological order described by the TQFTs in Eq. (5.20) is identical to that of the CR model in the oblique TI phase where a charge $(n, m)$ dyon condenses. The only remaining task is to explain the equivalence of their global symmetries, which is the topic we now turn to.

## 5.4 Global symmetries of the TQFT

In Section 5.2, we saw that the electric TQFT in Eq. (5.17) displays an emergent $\mathbb{Z}_{Nn}$ one-form symmetry,

$$a \to a + \frac{1}{Nn} \Upsilon, \qquad (5.27)$$

where $\Upsilon$ is a flat connection with quantized cycles, $\oint \Upsilon \in 2\pi\mathbb{Z}$. However, as in the discussion of the lattice model, this global symmetry acts on the electric loop operators in Eq. (5.22) as $\mathcal{O}_\Sigma \to \omega\, \mathcal{O}_\Sigma$, $\omega \in \mathbb{Z}_{Nn}$. In particular, the gauge invariant observables—the deconfined dyon loops of Eq. (5.24)—transform under this symmetry according to

$$\mathcal{D}_\Gamma \to \omega^{Nn/L}\, \mathcal{D}_\Gamma \,. \tag{5.28}$$

Such a transformation leaves the deconfined dyon loops invariant if $\omega \in \mathbb{Z}_{Nn/L} \subset \mathbb{Z}_{Nn}$. The remaining unbroken subgroup is then

$$G_{\text{oblique}} = \mathbb{Z}_{Nn/L} \,, \tag{5.29}$$

as we found in Section 4.2 in the context of the CR lattice model.

The same argument can be carried out for the magnetic $\mathbb{Z}_m$ one-form symmetry in Eq. (5.14), leading to a residual global symmetry, $\widetilde{G}_{\text{oblique}} = \mathbb{Z}_{m/L}$, acting on magnetic loops. However, we will demonstrate in the next section that this magnetic one-form symmetry has a mixed 't Hooft anomaly with $G_{\text{oblique}}$, and so we choose to view it as broken. Indeed, in the next section, we will consider the response of the oblique TI phase to the introduction of electric two-form probe fields. Finally, we remark that the theory also has a global discrete two-form symmetry [35, 60], but the presence of this symmetry will not play a role in the physics we describe in this work.

# 6   Generalized magnetoelectric effect

We now examine how the two-form probe field, $\mathcal{B}_{\mu\nu}$, introduced to the lattice model in Section 4.2, couples to the continuum TQFT. In the oblique TI phase, $\mathcal{B}_{\mu\nu}$ probes the emergent electric one-form symmetry, which we will demonstrate leads to a mixed 't Hooft anomaly with the emergent magnetic one-form symmetry. The coupling to $\mathcal{B}_{\mu\nu}$ ultimately leads to a higher-form generalization of the magnetoelectric effect, which provides a universal topological index characterizing an oblique confining phase.

We begin with the Cardy-Rabinovici action coupled to the probe, $\mathcal{B}_{\mu\nu}$, defined in Section 4.2. To determine how the probe couples to the effective field theory, we work in parallel to the arguments in Section 5.1. In the limit, $g^2 \to \infty$, and $\theta/2\pi = -n/m$, the

action becomes

$$
S[\mathcal{B}] = \sum_r \left[ -\frac{iNn}{4\pi m} \varepsilon_{\mu\nu\lambda\sigma} \left( \Delta_\mu a_\nu \right) \left( \Delta_\lambda a_\sigma \right) - \frac{iN\pi}{4m} \varepsilon_{\mu\nu\lambda\sigma} \left( s_{\mu\nu} + \frac{\mathcal{B}_{\mu\nu}}{N} \right) \left( s_{\lambda\sigma} + \frac{\mathcal{B}_{\lambda\sigma}}{N} \right) \right]
$$
$$
+ \frac{iNn}{m} \sum_r \left( m_\mu + \frac{1}{2} \varepsilon_{\mu\nu\lambda\sigma} \Delta_\nu \frac{\mathcal{B}_{\lambda\sigma}}{N} \right) a_\mu - iN \sum_\ell n_\mu a_\mu .
\tag{6.1}
$$

As in Section 5.1, we proceed by integrating out $a_\mu$, which leads to a local constraint that we can express in the action using a Lagrange multiplier, $\widetilde{\alpha}_\mu \in \mathbb{Z}$. The resulting action is

$$
S[\mathcal{B}] = \sum_r \left[ -\frac{2\pi i}{m} \frac{1}{2} \varepsilon_{\mu\nu\lambda\sigma} \left( s_{\mu\nu} + \frac{\mathcal{B}_{\mu\nu}}{N} \right) \Delta_\lambda \widetilde{\alpha}_\sigma - \frac{iNn\pi}{m} \frac{1}{4} \varepsilon_{\mu\nu\lambda\sigma} \left( s_{\mu\nu} + \frac{\mathcal{B}_{\mu\nu}}{N} \right) \left( s_{\lambda\sigma} + \frac{\mathcal{B}_{\lambda\sigma}}{N} \right) \right] ,
\tag{6.2}
$$

which reduces to Eq. (5.6) when $\mathcal{B}_{\mu\nu} = 0 \bmod N$, since $\mathcal{B}_{\mu\nu}$ can be absorbed into $s_{\mu\nu}$ in this case.

In contrast to the approach of Section 5.1, we will seek a dual theory in terms of "electric" variables prior to taking the continuum limit. We perform a duality transformation on the lattice by invoking the Poisson summation formula, which leads to the action,

$$
S[\mathcal{B}] = \sum_r \left[ -\frac{2\pi i}{m} \frac{1}{2} \varepsilon_{\mu\nu\lambda\sigma} \left( \xi_{\mu\nu} + \frac{\mathcal{B}_{\mu\nu}}{N} \right) \Delta_\lambda \zeta_\sigma - \frac{iNn\pi}{m} \frac{1}{4} \varepsilon_{\mu\nu\lambda\sigma} \left( \xi_{\mu\nu} + \frac{\mathcal{B}_{\mu\nu}}{N} \right) \left( \xi_{\lambda\sigma} + \frac{\mathcal{B}_{\lambda\sigma}}{N} \right) \right.
$$
$$
\left. -2\pi i \frac{1}{4} \varepsilon_{\mu\nu\lambda\sigma} \xi_{\mu\nu} t_{\lambda\sigma} - 2\pi i \, \zeta_\mu \rho_\mu \right] ,
\tag{6.3}
$$

where we have introduced $\xi_{\mu\nu}, \zeta_\mu \in \mathbb{R}$ and $t_{\mu\nu}, \rho_\mu \in \mathbb{Z}$, which are all dynamical. Integrating over $\xi_{\mu\nu}$ gives

$$
S[\mathcal{B}] = \sum_r \left[ \frac{2\pi i}{Nn} \frac{1}{2} \varepsilon_{\mu\nu\lambda\sigma} t_{\mu\nu} \Delta_\lambda \zeta_\sigma + \frac{2\pi i}{N} \frac{1}{4} \varepsilon_{\mu\nu\lambda\sigma} t_{\mu\nu} \mathcal{B}_{\lambda\sigma} + \frac{im\pi}{Nn} \frac{1}{4} \varepsilon_{\mu\nu\lambda\sigma} t_{\mu\nu} t_{\lambda\sigma} - 2\pi i \, \zeta_\mu \rho_\mu \right.
$$
$$
\left. + \frac{i\pi}{Nnm} \varepsilon_{\mu\nu\lambda\sigma} \left( \Delta_\mu \zeta_\nu \right) \left( \Delta_\lambda \zeta_\sigma \right) \right] ,
\tag{6.4}
$$

and integrating over $\zeta_\mu \in \mathbb{R}$ implies a constraint on $t_{\mu\nu}$,

$$
\frac{1}{2} \varepsilon_{\mu\nu\lambda\sigma} \Delta_\nu t_{\lambda\sigma} \in Nn \, \mathbb{Z}.
\tag{6.5}
$$

We then introduce a Lagrange multiplier, $\alpha_\mu \in \mathbb{Z}$, for this constraint and arrive at the action,

$$
S[\mathcal{B}] = \sum_r \left[ \frac{2\pi i}{Nn} \frac{1}{2} \varepsilon_{\mu\nu\lambda\sigma} t_{\mu\nu} \Delta_\lambda \alpha_\sigma + \frac{2\pi i}{N} \frac{1}{4} \varepsilon_{\mu\nu\lambda\sigma} t_{\mu\nu} \mathcal{B}_{\lambda\sigma} + \frac{im\pi}{Nn} \frac{1}{4} \varepsilon_{\mu\nu\lambda\sigma} t_{\mu\nu} t_{\lambda\sigma} \right] .
\tag{6.6}
$$

This action (with $\mathcal{B}_{\mu\nu} = 0 \bmod N$) is the "electric" analogue of Eq. (5.6).

We may use this result to construct a corresponding continuum field theory. First, consider the case when $\mathcal{B}_{\mu\nu} = 0 \bmod N$. Eq. (6.6) has emergent zero-form and one-form gauge symmetries,

$$\alpha_\mu \to \alpha_\mu - m\,\eta_\mu + \Delta_\mu \chi + Nn\,\mathcal{N}_\mu\,, \qquad t_{\mu\nu} \to t_{\mu\nu} + (\Delta_\mu\,\eta_\nu - \Delta_\nu\,\eta_\mu) + Nn\,\mathcal{M}_{\mu\nu}\,, \quad (6.7)$$

where $\chi, \mathcal{N}_\mu \in \mathbb{Z}$ are the parameters for zero-form gauge transformations and $\eta_\mu, \mathcal{M}_{\mu\nu} \in \mathbb{Z}$ are independent parameters for one-form gauge transformations. The same reasoning from Section 5.1 then suggests the correspondence,

$$\frac{2\pi}{Nn}\alpha_\mu \to a_\mu, \qquad \frac{2\pi}{Nn}t_{\mu\nu} \to b_{\mu\nu}, \quad (6.8)$$

implying that the continuum analogue of Eq. (6.6) (with $\mathcal{B}_{\mu\nu} = 0$) is the TQFT in Eq. (5.17), consistent with our arguments in the continuum. We recall from Section 5.2.2 that the $U(1)$ one-form field, $a$, is the electric gauge field, and the $U(1)$ two-form field, $b$, represents world sheets of flux tubes.

We now consider how to treat the case with the probe turned on, $\mathcal{B}_{\mu\nu} \neq 0 \bmod N$. We observe that Eq. (6.6) is invariant under one-form gauge transformations for $\mathcal{B}_{\mu\nu}$,

$$\mathcal{B}_{\mu\nu} \to \mathcal{B}_{\mu\nu} + \Delta_\mu\eta_\nu - \Delta_\nu\eta_\mu, \qquad \alpha_\mu \to \alpha_\mu + n\,\eta_\mu, \quad (6.9)$$

where $\eta_\mu \in \mathbb{Z}$. Although we use the same notation, note that this gauge transformation is independent of the transformation in Eq. (6.7). The correspondence in Eq. (6.8) suggests that the gauge symmetry for $\mathcal{B}_{\mu\nu}$ becomes a $\mathbb{Z}_N$ gauge symmetry in the TQFT. Thus, in the continuum, $\mathcal{B}_{\mu\nu}$ corresponds to a background $\mathbb{Z}_N$ two-form gauge field. We then make the replacement,

$$\frac{2\pi\mathcal{B}_{\mu\nu}}{N} \to B_{\mu\nu}\,. \quad (6.10)$$

To guarantee that $B_{\mu\nu}$ is a $\mathbb{Z}_N$ gauge field, we introduce a new charge-$N$ Lagrange multiplier field, $\beta_\mu$, in the continuum theory to Higgs $B$. Putting everything together, we may write the continuum TQFT as

$$S[B] = \frac{iNn}{2\pi} \int b \wedge (da + B) + \frac{iNnm}{4\pi} \int b \wedge b + \frac{iN}{2\pi} \int B \wedge d\beta\,. \quad (6.11)$$

Integrating over $\beta$ enforces a constraint that $dB = 0$ locally and

$$\oint_\Sigma B \in \frac{2\pi}{N}\,\mathbb{Z} \quad (6.12)$$

for any closed surface $\Sigma$. We also comment that Eq. (6.11) is invariant under the one-form gauge symmetry,

$$b \to b + d\lambda, \qquad a \to a - m\,\lambda, \qquad \beta \to \beta - n\,\lambda, \tag{6.13}$$

which is the continuum analogue of Eq. (6.7). Here, $\lambda$ is a $U(1)$ one-form gauge field.

We now describe the physical interpretation of Eq. (6.11) in terms of the global one-form symmetries. We note that the action, Eq. (6.11), is invariant mod $2\pi i$ under

$$B \to B + d\lambda, \qquad a \to a - \lambda, \tag{6.14}$$

where $\lambda$ is a $U(1)$ one-form gauge field. This one-form gauge symmetry for $B$ is the continuum analogue of Eq. (6.9) and is independent of the transformation in Eq. (6.13). Hence, $B$ is a background gauge field for the emergent global electric one-form symmetry *of the TQFT*. More precisely, since $B$ is a $\mathbb{Z}_N$ gauge field (after integrating out $\beta$), it probes a subgroup $\mathbb{Z}_N \subset \mathbb{Z}_{Nn}$ of the global electric one-form symmetry described in Section 5.2.2.

We now consider the effect of $B_{\mu\nu}$ on the magnetic one-form symmetry, which we recall is $\mathbb{Z}_m$ before being broken to a subgroup by oblique confinement. We will see that the global magnetic one-form symmetry of the TQFT is broken by $B$, thus demonstrating that there is a mixed 't Hooft anomaly between the electric and magnetic one-form symmetries. The magnetic one-form symmetry is easiest to view when the theory is presented in terms of the magnetic fields, $\tilde{a}$ and $\tilde{b}$, as in Eq. (5.11). To determine how $B$ couples to the magnetic version of the theory, we then complete a standard electromagnetic duality calculation and perform a Hubbard-Stratonovich transformation, which introduces the two-form field, $\tilde{b}$. The result is

$$\widetilde{S}[B] = -\frac{im}{2\pi} \int \tilde{b} \wedge d\tilde{a} - \frac{iNnm}{4\pi} \int \tilde{b} \wedge \tilde{b} + \frac{i}{2\pi} \int B \wedge d\tilde{a} + \frac{iN}{2\pi} \int B \wedge d\beta, \tag{6.15}$$

where $\tilde{a}$ and $\tilde{b}$ are the magnetic counterparts of $a$ and $b$, introduced in Section 5.2. We remark that in the magnetic varibales of Eq. (6.15), the loop variables—and thus the global electric one-form symmetry—have been fractionalized [62], i.e. in the magnetic representation the surface operators, $\exp\left(\oint \tilde{b}\right)$, carry fractional charge. This phenomenon does not occur in the electric representation of the theory, Eq. (6.11), and therefore is not essential to understanding the topological order of the oblique TI. It is physically a consequence of the fact that dyon loops are composites of electric and magnetic loops.

It is clear now that the magnetic one-form symmetry is broken. Under the transformation for the global $\mathbb{Z}_m$ magnetic one-form symmetry, Eq. (5.14), the action, Eq. (6.15), changes by

$$\delta_\Upsilon \widetilde{S}[B] = -\frac{2\pi i}{m} \int \frac{dB}{2\pi} \wedge \frac{\Upsilon}{2\pi} \in \frac{2\pi i}{m} \mathbb{Z}, \tag{6.16}$$

so the partition function is not left invariant. Therefore, the background field, $B$, has broken the magnetic one-form symmetry. Because $B$ is a background gauge field for the electric one-form symmetry of the TQFT, we conclude that there is a mixed 't Hooft anomaly for the electric and magnetic one-form symmetries of the TQFT, as stressed in previous sections.

We can now observe that, in an oblique TI, the coupling to $B$ leads also to a universal "response" generalizing the magnetoelectric effect of ordinary TIs. After integrating out $b$ in Eq. (6.11), the resulting effective action is

$$S_{\text{eff}}[B] = -\frac{iNn}{4\pi m} \int (da + B) \wedge (da + B) + \frac{iN}{2\pi} \int B \wedge d\beta \,. \tag{6.17}$$

Typically, in calculating response, one integrates out all fluctuating fields, but we cannot further integrate out $a$ since it is required for one-form gauge invariance under Eq. (6.14). However, we may nonetheless eliminate $a$ by gauge fixing $B$, $B \to B - da$, in analogy with fixing to unitary gauge in a Laudau-Ginzburg theory of a superconductor. The resulting response theory is then

$$S_{\text{response}}[B] = -\frac{iNn}{4\pi m} \int B \wedge B \,, \tag{6.18}$$

where integrating over $\beta$ fixes $B$ to be a $\mathbb{Z}_N$ field.

Because $B$ does not couple to the global current for a continuous symmetry (the global charge is not conserved except mod $Nn/L$), we emphasize that the notion of the generalized magnetoelectric effect as a genuine response coefficient does not have a straightforward physical interpretation, unlike its $U(1)$ counterpart [4]. In addition, one is technically allowed to eliminate $a$ by gauge fixing $B$ only if the phase is topologically trivial (i.e., if $L = 1$). Nevertheless, the correct interpretation of the fractional coefficient in Eqs. (6.17) – (6.18) is as a universal topological index for an oblique confining phase, representing a higher-form generalization of the magnetoelectric effect. In particular, because this generalized magnetoelectric effect probes the residual one-form symmetry, $G_{\text{oblique}} = \mathbb{Z}_{Nn/L}$, and is therefore sensitive to the electric and magnetic charges condensed in an oblique TI phase, its presence distinguishes the oblique TI from an ordinary $\mathbb{Z}_L$ topological order.

Finally, we also comment on an ambiguity in the response computed using the TQFT— states with the same bulk topological order may have a different response coefficient. Integrating out $a$ in Eq. (6.11) quantizes the cycles of $b$ such that the partition function is invariant under $m \to m + 2Nn$, and indeed, from our discussion of the operators and correlation functions in Section 5.3, one can easily verify that the bulk topological order is invariant under $m \to m + 2Nn$. A similar ambiguity appears in the fractional quantum Hall

effect, where the effective Chern-Simons theory only determines the fractional part of the Hall conductivity since one can always "add Landau levels". We note, however, that for a given $N$, oblique confining states with different topological orders will never have the same response coefficient in Eq. (6.18), so in this sense, the coefficient is a universal topological index that characterizes the state. We also remark that there no ambiguity in the response when the theory is on a manifold with a boundary. In the fractional quantum Hall problem, a boundary is required to inject a current and measure the Hall voltage, though the Hall conductivity is still a feature characterizing local response of the quantum Hall fluid regardless of whether there is a boundary. The same considerations should apply here to the response for an oblique confining state.

# 7 Gapped boundary states of oblique TIs

## 7.1 Boundary topological order and anomaly inflow

As we have seen, the bulk of an oblique confining phase is topologically ordered in general, so it is natural to consider the theory defined on an open manifold and investigate its possible boundary states. We will find that the boundary of an oblique TI phase hosts a topological order not realizable in a purely 2+1-D system. However, we will also show that the choice of boundary topological order is not uniquely determined by the bulk theory, and we will explicitly construct a new boundary state distinct from those discussed in the past for TQFTs of the type in Eq. (5.17). We will also see that these different possibilities for surface topological order exhibit breaking of different subgroups of the emergent bulk one-form symmetry, and as such are intimately linked to the presence of this global symmetry in a manner reminiscent of ordinary symmetry protection in SPT and SET phases.

The first boundary state we will consider was previously studied in Refs. [35, 38, 60] and is given by the action

$$S = \int_X \left( \frac{iNn}{2\pi} b \wedge da + \frac{iNnm}{4\pi} b \wedge b \right) + \int_{\partial X} \left( -\frac{iNnm}{4\pi} c \wedge dc + \frac{iNn}{2\pi} c \wedge da \right), \quad (7.1)$$

where $c$ is a one-form $U(1)$ gauge field that exists solely on the boundary of the manifold $X$. We review this analysis and then construct the new topologically ordered boundary state,

$$\widetilde{S} = \int_X \left( -\frac{iNn}{2\pi} db \wedge a + \frac{iNnm}{4\pi} b \wedge b \right) + \int_{\partial X} \left( \frac{iNnm}{4\pi} \tilde{c} \wedge d\tilde{c} + \frac{iNn}{2\pi} b \wedge (m\tilde{c} - d\phi) \right), \quad (7.2)$$

where $\tilde{c}$ is a one-form $U(1)$ gauge field and $\phi$ is a $2\pi$-periodic scalar field, both defined only on $\partial X$. We refer to the boundary state in Eq. (7.1) as the "electric boundary condition",

as it describes a boundary topological order with $|Nn|$ anyons, and we call Eq. (7.2) the "magnetic boundary condition", which hosts a topological order with $|m|$ anyons. The remainder of this section will concern the construction of these boundary states and their physical consequences.

### 7.1.1 Electric boundary condition

We begin by reviewing the boundary state introduced in Refs. [35, 38, 60]. This boundary state is equivalent to taking the boundary condition $b|_{\partial X} = 0$, but here we give an argument of anomaly inflow for the one-form gauge symmetry. If the theory, Eq. (5.17), is on a manifold $X$ that has a boundary, then the action in Eq. (5.17) is no longer invariant under one-form gauge transformations,

$$a \to a - m\,\lambda, \qquad b \to b + d\lambda. \tag{7.3}$$

Under a one-form gauge transformation, the bulk action, Eq. (5.17), changes by the boundary term,

$$\delta_\lambda S_{\text{bulk}} = -\frac{iNnm}{4\pi} \int_{\partial X} \lambda\, d\lambda + \frac{iNn}{2\pi} \int_{\partial X} \lambda\, da. \tag{7.4}$$

One can introduce a one-form gauge field $c$ that exists solely on the boundary and transforms under one-form gauge transformations as

$$c \to c - \lambda. \tag{7.5}$$

If we take the boundary action to be

$$S_{\text{bdry}} = -\frac{iNnm}{4\pi} \int_{\partial X} c\, dc + \frac{iNn}{2\pi} \int_{\partial X} c\, da, \tag{7.6}$$

then the total action for the bulk and boundary together, given by Eq. (7.1), is gauge invariant. In addition, since this action is invariant under Eq. (5.19), we see that the global $\mathbb{Z}_{Nn}$ electric one-form symmetry is preserved.

To determine the physics of this boundary state, we construct the gauge invariant observables. Two such operators to consider are

$$\begin{aligned}
\widetilde{\mathcal{U}}_\Sigma &= \exp\left( i \oint_\Gamma c + i \int_\Sigma b \right), \\
V_\Gamma &= \exp\left( i \oint_\Gamma a - im \oint_\Gamma c \right),
\end{aligned} \tag{7.7}$$

where $\Gamma = \partial\Sigma$ is a curve that lies in $\partial X$, and the surface $\Sigma$ can be in the bulk. The equation of motion for $a$ in the bulk ensures that $db = 0$ locally while globally we have

$$\oint b \in \frac{2\pi}{Nn}\mathbb{Z}, \tag{7.8}$$

so $b$ is a $\mathbb{Z}_{Nn}$ gauge field. Therefore, physical loop operators constructed from $\widetilde{\mathcal{U}}_\Sigma$ are generated by

$$\widetilde{V}_\Gamma = \left(\widetilde{\mathcal{U}}_\Sigma\right)^{Nn}. \tag{7.9}$$

However, the equation of motion for $a$ on $\partial X$ constrains the value of $b$ at the surface to be

$$b|_{\partial X} = -dc, \tag{7.10}$$

so $\widetilde{V}_\Gamma$ is actually a trivial operator, which indicates that this boundary state is equivalent to the boundary condition $b|_{\partial X} = 0$. Thus, since $\widetilde{V}_\Gamma$ is trivial, all nontrivial loop operators on the surface are generated by $V_\Gamma$. Using the boundary action, the correlation functions are given by

$$\left\langle (V_\Gamma)^k \, (V_{\Gamma'})^{k'} \right\rangle = \exp\left(\frac{2\pi i \, kk'm}{Nn} \varphi_{\text{link}}[\Gamma, \Gamma']\right), \tag{7.11}$$

where $\varphi_{\text{link}}[\Gamma, \Gamma']$ is the linking number of the closed loops $\Gamma$ and $\Gamma'$. Thus, the loop operators generated $V_\Gamma$ represent the world lines of anyons on the boundary $\partial X$. Since $V_\Gamma$ generates the deconfined particles on $\partial X$ and transforms under the global electric one-form symmetry, Eq. (5.19), we see that the topological order on $\partial X$ is realized by completely breaking the global $\mathbb{Z}_{Nn}$ electric one-form symmetry.

Next, we count the types of anyons in the surface topological order. For simplicity, let us assume that $Nn > 0$. From Eq. (7.11), we observe that for $0 \le k < \frac{Nn}{L}$, the operator $(V_\Gamma)^k$ is an anyon and is hence restricted to the boundary, $\partial X$. But $(V_\Gamma)^{Nn/L}$ is equivalent to $\mathcal{D}_\Gamma$, a loop operator of the bulk (see Section 5.3). Therefore, $\mathcal{D}_\Gamma = (V_\Gamma)^{Nn/L}$ can freely move into the bulk and thus must be either a boson or fermion, as reflected in the correlation functions, Eq. (7.11). The remaining loop operators on the boundary represent anyons resulting from fusing a bulk quasiparticle and an anyon $(V_\Gamma)^k$ with $0 \le k < \frac{Nn}{L}$. Since there are $L$ bulk loop operators, the total number of boundary loop operators is $\frac{Nn}{L} \cdot L = Nn$. This boundary theory has fewer observables than one would expect for a theory with the action in Eq. (7.6) solely in 2+1-D. The reason is that we require the observables to be invariant under the one-form gauge symmetry in Eqs. (7.3) and (7.5).

As we have discussed, when $L > 1$, the bulk is topologically ordered and hosts nontrivial deconfined quasiparticles, $\mathcal{D}_\Gamma = (V_\Gamma)^{Nn/L}$, which have trivial mutual statistics with all anyons

of the boundary theory. Since these bulk particles may also be obtained by fusing boundary anyons, the boundary theory is not consistent as a purely 2+1-D topological order because it is non-modular. A topological order of this type cannot exist purely in 2+1-D but must necessarily be coupled to a 3+1-D bulk.

### 7.1.2 Magnetic boundary condition

We now turn to the second boundary state. This state is equivalent to taking the boundary condition $a|_{\partial X} = 0$, though we will opt for a gauge invariant description as we did for the first boundary state. In Section 7.1.1, we chose to write the BF term in the bulk action as $b \wedge da$ and then introduced boundary degrees of freedom that ensured gauge invariance under one-form gauge transformations, Eq. (7.3). Now, we construct an alternate boundary state by writing the BF term in the form $-db \wedge a$ and examining how the bulk action changes under a gauge transformation. Of course, if $X$ has no boundary, then these two ways of writing the BF term are equivalent, but when $X$ has a boundary, we will obtain different physics. We thus take the bulk action to be

$$\widetilde{S}_{\text{bulk}} = \int_X \left( -\frac{iNn}{2\pi} db \wedge a + \frac{iNnm}{4\pi} b \wedge b \right). \tag{7.12}$$

We consider zero-form and one-form gauge transformations, given by

$$b \to b + d\lambda, \qquad a \to a - m\,\lambda + d\xi, \tag{7.13}$$

where $\lambda$ is a $U(1)$ one-form gauge field and $\xi$ is a $2\pi$-periodic scalar. Under these gauge transformations, $\widetilde{S}_{\text{bulk}}$ changes by the boundary term,

$$\delta_\lambda \widetilde{S}_{\text{bulk}} = \int_{\partial X} \left( \frac{iNnm}{4\pi} \lambda \wedge d\lambda + \frac{iNnm}{2\pi} b \wedge \lambda - \frac{iNn}{2\pi} b \wedge d\xi \right). \tag{7.14}$$

We then introduce a one-form gauge field $\tilde{c}$ and a $2\pi$-periodic scalar field $\phi$ that exist solely on $\partial X$ and transform under the gauge symmetry of Eq. (7.13) by

$$\tilde{c} \to \tilde{c} - \lambda, \qquad \phi \to \phi - \xi. \tag{7.15}$$

If we take the boundary action to be

$$\widetilde{S}_{\text{bdry}} = \int_{\partial X} \left( \frac{iNnm}{4\pi} \tilde{c} \wedge d\tilde{c} + \frac{iNnm}{2\pi} \tilde{c} \wedge b - \frac{iNn}{2\pi} b \wedge d\phi \right), \tag{7.16}$$

then the total action, Eq. (7.2), is fully gauge invariant under Eqs. (7.13) and (7.15). The theory is also invariant under zero-form gauge transformations for $\tilde{c}$,

$$\tilde{c} \to \tilde{c} + d\chi, \qquad a \to a - m\,d\chi, \tag{7.17}$$

where $\chi$ is a compact scalar. However, note that the action now is not invariant under Eq. (5.19), so the $\mathbb{Z}_{Nn}$ electric one-form symmetry is explicitly broken by this boundary state.

We then determine the topological order of this second boundary state, again by finding the gauge invariant observables and computing their correlation functions. One gauge invariant operator is

$$\widehat{V}_\Gamma = \exp\left( i\phi(\mathcal{P}) - i\phi(\mathcal{P}') + i \int_\Gamma a - im \int_\Gamma \tilde{c} \right), \tag{7.18}$$

where $\mathcal{P}$ and $\mathcal{P}'$ are points on $\partial X$, and $\Gamma$ is a curve in $\partial X$ with endpoints at $\mathcal{P}$ and $\mathcal{P}'$. Additionally, we can take $\mathcal{P} = \mathcal{P}'$ and consider a closed loop $\Gamma$. However, the equation of motion for $b$ renders $\widehat{V}_\Gamma$ trivial[12]. The only other gauge invariant operator that can be constructed with the fields on $\partial X$ is

$$\widetilde{\mathcal{U}}_\Sigma = \exp\left( i \oint_\Gamma \tilde{c} + i \int_\Sigma b \right), \tag{7.19}$$

where $\Gamma = \partial\Sigma$ lies on the boundary $\partial X$. Integrating out $a$ constrains $b$ to be a $\mathbb{Z}_{Nn}$ gauge field in the bulk, and $\phi$ imposes this restriction on the boundary. Thus, the physical loop operators on the boundary are generated by

$$\widetilde{V}_\Gamma = \left( \widetilde{\mathcal{U}}_\Sigma \right)^{Nn}, \tag{7.20}$$

which have correlation functions of

$$\left\langle \left( \widetilde{V}_\Gamma \right)^k \left( \widetilde{V}_{\Gamma'} \right)^{k'} \right\rangle = \exp\left( -\frac{2\pi i\, kk'Nn}{m} \varphi_{\text{link}}[\Gamma, \Gamma'] \right), \tag{7.21}$$

where $\varphi_{\text{link}}[\Gamma, \Gamma']$ is the linking number introduced in Eq. (7.11). Hence, the operators $(\widetilde{V}_\Gamma)^k$ represent anyons, and Eq. (7.21) describes their fractional statistics.

Let us now count how many particles participate in this surface topological order. As in the analysis above, we take $m > 0$ for simplicity. Then, the correlation functions in Eq. (7.21) tell us that $(\widetilde{V}_\Gamma)^k$, with $0 \le k < \frac{m}{L}$, is an anyon confined to the boundary. But $(\widetilde{V}_\Gamma)^{m/L}$ is equivalent to the bulk particle $\mathcal{D}_\Gamma$. The remaining anyons are formed by fusing a bulk particle with $(\widetilde{V}_\Gamma)^k$, where $0 \le k < \frac{m}{L}$. Therefore, the number of anyons in this surface topological order is $m$.

We have thus found an alternate boundary state characterized by a different topological order. In particular, the topological data for this boundary state can be obtained by taking $(Nn, m) \to (m, -Nn)$ for the topological data computed for the first boundary state.

---

[12]Since the loop operators of $a$ are trivial on $\partial X$, this boundary state is equivalent to the boundary condition $a|_{\partial X} = 0$.

### 7.1.3   Global symmetries of the boundary theories

We now comment in more detail on the realizations of the emergent bulk global symmetries in each boundary state. We first discuss the emergent $\mathbb{Z}_{Nn}$ electric one-form symmetry that we have primarily focused on thus far. The "electric" boundary state, Eq. (7.1), has an action that respects the global $\mathbb{Z}_{Nn}$ one-form symmetry of the bulk, but this symmetry is ultimately broken completely in the IR by the presence of $|Nn|$ deconfined anyons on the surface. On the other hand, this $\mathbb{Z}_{Nn}$ one-form symmetry is explicitly broken by the "magnetic" boundary state, Eq. (7.2).

The global $\mathbb{Z}_m$ magnetic one-form symmetry, Eq. (5.14)—which we recall has a mixed anomaly with the $\mathbb{Z}_{Nn}$ electric symmetry—has the opposite fate. This can be most clearly understood by acting with electromagnetic duality on these boundary states, which we do in Appendix D, so that they are constructed in terms of the magnetic gauge fields, $\tilde{a}$ and $\tilde{b}$. Indeed, the electric boundary state, Eq. (7.1), has an action that explicitly breaks the magnetic one-form symmetry, and the action of the magnetic boundary state, Eq. (7.2), preserves the magnetic one-form symmetry at the level of the action, but its $|m|$ deconfined anyons are charged under this global symmetry. The boundary topological order in the magnetic boundary state is thus realized by breaking the magnetic one-form symmetry completely.

Interestingly, the presence of an unbroken subgroup of the global one-form symmetry in the bulk has played an essential role in this analysis. For example, although the electric $\mathbb{Z}_{Nn}$ one-form symmetry is broken down to $G_{\mathrm{oblique}} = \mathbb{Z}_{Nn/L}$ by the bulk topological order (i.e. at the level of the Hilbert space), the existence of the full emergent $\mathbb{Z}_{Nn}$ symmetry determines the boundary topological order under the electric boundary conditions. The same can be said of the relationship between the full emergent $\mathbb{Z}_m$ magnetic one-form symmetry and the boundary state obtained using magnetic boundary conditions.

## 7.2   Boundary Hall conductivity

In the previous subsection, we constructed two different boundary states, which are topologically ordered. An important piece of data characterizing these topological orders is the Hall conductivity, which we now compute. We begin with the first boundary state, Eq. (7.6), which we couple to a background $U(1)$ gauge field, $A_\mu$. To ensure invariance under the one-form gauge symmetry, this background field must couple to $(da - m\,dc)$,

$$S_{\mathrm{bdry}}[A] = \int_{\partial X} \left( -\frac{iNnm}{4\pi} c\,dc + \frac{iNn}{2\pi} c\,da + \frac{i}{2\pi} A\,(da - m\,dc) \right). \qquad (7.22)$$

The physical interpretation of this particular coupling is that the charge carriers are dyons. If we integrate out $a$ and $c$ to obtain an effective action for $A$, we observe that the Hall conductivity for $A$ is

$$\sigma_H = \frac{m}{Nn} \qquad (7.23)$$

in units of $e^2/h$.

We perform a similar calculation for the second boundary state. First, note that integrating over $\phi$ gives $b|_{\partial X} = d\tilde{a}/Nn$ where $\tilde{a}$ is a $U(1)$ gauge field. Thus, the boundary action, Eq. (7.16), becomes

$$\widetilde{S}_{\text{bdry}} = \int_{\partial X} \left( \frac{iNnm}{4\pi} \tilde{c}\, d\tilde{c} + \frac{im}{2\pi} \tilde{c}\, d\tilde{a} \right). \qquad (7.24)$$

Because of one-form gauge invariance, the background field $A$ couples to $Nn(b + d\tilde{c}) = d\tilde{a} + Nn\, d\tilde{c}$, which gives us an action of

$$\widetilde{S}_{\text{bdry}}[A] = \int_{\partial X} \left( \frac{iNnm}{4\pi} \tilde{c}\, d\tilde{c} + \frac{im}{2\pi} \tilde{c}\, d\tilde{a} - \frac{i}{2\pi} A(d\tilde{a} + Nn\, d\tilde{c}) \right). \qquad (7.25)$$

After calculating the effective action for $A$, we find a Hall conductivity of

$$\widetilde{\sigma}_H = \frac{Nn}{m}, \qquad (7.26)$$

which is similar to the Hall conductivity for the first boundary state except that the electric charge $Nn$ and magnetic charge $m$ have been swapped.

# 8  Comments on duality and modular invariance

Before concluding, we wish to comment in more detail on the relationship between the duality transformations of Cardy and Rabinovici (CR), which we introduced in Section 3.3, and the duality transformations invoked in studying the TQFT in Section 5. Recall from Section 3.3 that, in the Coulomb gas picture, the free energy of the CR model is invariant under

$$\begin{aligned} \mathcal{S}_{\text{CR}} &: \quad (n, m) \mapsto (-m, n), \\ \mathcal{T}_{\text{CR}} &: \quad (n, m) \mapsto (n - m, m). \end{aligned} \qquad (8.1)$$

Although these transformations leave the free energy invariant, they do not preserve the correlation functions of the theory and thus the topological order [33], which are determined by $L = \gcd(Nn, m)$. Under the CR modular transformations, $L$ is clearly preserved by $\mathcal{T}_{\text{CR}}$ but not by $\mathcal{S}_{\text{CR}}$. As a result, these modular transformations map the different oblique TI phases of the CR model to one another, generating the phase diagram in Figure 4. We

therefore caution that while $\mathcal{S}_{\mathrm{CR}}$ is often referred to as "electromagnetic duality," it does not preserve the ground state observables of the theory[13]. We note also that the generalized magnetoelectric effect, Eq. (6.17), transforms under both of these transformations.

On the other hand, it is possible to write down a set of modular transformations that preserves $L$ and, therefore, the topological order and global symmetries of an oblique TI phase,

$$
\begin{aligned}
\mathcal{S} : & \quad (Nn, m) \mapsto (m, -Nn), \\
\mathcal{T} : & \quad (Nn, m) \mapsto (Nn - m, m).
\end{aligned}
\tag{8.2}
$$

The $\mathcal{S}$ transformation is already familiar to us from Eq. (5.20) as the physical electromagnetic (EM) duality transformation, which exchanges the total electric and magnetic charges comprising the condensing dyons in a particular oblique TI phase. The transformation, $\mathcal{T}$, is a shift of the *total* $\Theta$-angle, $\Theta = -2\pi N\, n/m \to \Theta + 2\pi$, which also appears to be a symmetry of the partition function. However, there is a well-known subtlety to this analysis. Namely, one must also consider the effects of $\mathcal{S}$ and $\mathcal{T}$ on the statistics of point excitations. For the CR model coupled to microscopic fermions (which involes a spin$_c$ connection), the $\mathcal{T}$ transformation preserves the statistics of point excitations, but EM duality, $\mathcal{S}$, does not. Nevertheless, there is an alternate notion of electromagnetic duality for fermions, which we discuss in Appendix C. For the bosonic CR model (with ordinary $U(1)$ gauge fields), $\mathcal{S}$ manifestly preserves the topological order, but $\mathcal{T}$ can transmute statistics of the point excitations. A shift of $\Theta$ by $4\pi$, i.e. $\mathcal{T}^2$, on the other hand, leaves the statistics invariant [8, 9].

The above statements about invariance under periodicity, $\mathcal{T}$, are transparent at the level of the bulk field theory. Consider the expression for the bosonic bulk TQFT in the "magnetic" variables,

$$
\widetilde{S} = -\frac{im}{2\pi} \int \tilde{b} \wedge d\tilde{a} - \frac{iNnm}{4\pi} \int \tilde{b} \wedge \tilde{b}.
\tag{8.3}
$$

Integrating out $\tilde{a}$ constrains $\tilde{b}$ to be a $\mathbb{Z}_m$ gauge field. Thus, after integrating out $\tilde{a}$, what remains is a topological term that on a closed manifold, $Y$, evaluates to

$$
\widetilde{S}_{\mathrm{eff}} = -\frac{iNnm}{4\pi} \int_Y \tilde{b} \wedge \tilde{b} \in \frac{\pi i Nn}{m} \mathbb{Z}.
\tag{8.4}
$$

We thus conclude that $Nn \sim Nn - 2m$, meaning that $\mathcal{T}^2$ leaves the partition function

---

[13]In the absence of a $\Theta$-angle, the duality of ordinary $\mathbb{Z}_N$ gauge theories [46, 47] is equivalent to EM duality in the sense used by Cardy and Rabinovici in Refs. [23, 24], in that it amounts to the exchange of electric and magnetic charges. However, as we have shown, if $\Theta \neq 0$ the mapping is more subtle and it is incorrect to view the CR transformation as "electromagnetic" duality.

invariant, as anticipated above. In Appendix C, we similarly show that $\mathcal{T}$ leaves the partition function invariant in the fermionic case.

Finally, it is important to note how $\mathcal{S}$ and $\mathcal{T}$ act on the generalized magnetoelectric response, Eq. (6.17). Because EM duality, $\mathcal{S}$, is a genuine duality transformation of the theory that is simply an exact change of variables in the partition function, it preserves the generalized magnetoelectric effect. This can be seen by shifting $\tilde{b}$ by $B/m$ in Eq. (6.15) and inspecting the coefficient of $B \wedge B$. Put differently, even though $\mathcal{S}$ alters the form of the coupling to $B$ (the theory is not self-dual), $B$ still leads to the same universal index, which reflects the mixed anomaly between the electric and magnetic one-form symmetries. In contrast, $\mathcal{T}$ *does not* preserve the generalized magnetoelectric effect, since the terms involving the background field in Eq. (6.17) are not periodic.

To summarize, the modular transformations, $\mathcal{S}$ and $\mathcal{T}$, have a different physical meaning from the Cardy-Rabinovici transformations, $\mathcal{S}_{\mathrm{CR}}$ and $\mathcal{T}_{\mathrm{CR}}$. In the bosonic CR model, although $(\mathcal{T}_{\mathrm{CR}})^2 = \mathcal{T}^{2N}$ preserves the topological order, $\mathcal{S}_{\mathrm{CR}}$ does not in general. The CR modular transformations, $\mathcal{S}_{\mathrm{CR}}$ and $\mathcal{T}_{\mathrm{CR}}$, are symmetries of the free energy and map between the different phases of the theory, whereas $\mathcal{S}$ and $\mathcal{T}$ are genuine EM duality transformations that preserve correlation functions and thus the topological order.

## 9 Discussion

We have presented in this work a new class of 3+1-D fractional topological insulator phases based on oblique confinement and dyon condensation, using the Cardy-Rabinovici lattice gauge theories as foundational examples. These oblique topological insulator phases are characterized by both topological order and emergent one-form symmetries that remain unbroken in the infrared. We have shown that they exhibit exotic behavior such as gapped boundary topological orders not realizable strictly in 2+1-D, as well as a generalization of the magnetoelectric effect in the presence of two-form probe fields. Taken together, we believe these features motivate a new organizing paradigm for 3+1-D FTI phases, as well as SET phases more broadly, where the presence of unbroken emergent higher-form symmetries in the IR plays an essential role.

We comment that the oblique TI phases discussed in this work all correspond to ground states of Walker-Wang models [15–17, 54] or their generalizations with additional fundamental fermions [63, 64], which also should have the same emergent global symmetries, boundary states, and generalized magnetoelectric effect. The utility of the Cardy-Rabinovici model, however, is to provide a single unifying parent theory for each of these states and transitions

between them, all based on the physical mechanism of oblique confinement. In the future, it would be very interesting to search for models leading to oblique TI phases that cannot be represented using Walker-Wang models.

Because we have constructed two distinct boundary topological orders for the same oblique TI bulk, which differ in the emergent one-form symmetries they break in the IR, it is natural to wonder if a continuous quantum phase transition between them is possible. Any such transition would correspond to an additional gapless boundary state sharing a gauge anomaly with the bulk, and it would constitute a new type of phase transition beyond the Landau ordering framework.

While our focus in this work was on bulk $\mathbb{Z}_N$ gauge theories, it should be possible to construct more general classes of oblique TI phases. For example, one can consider possible phases in which the loop degrees of freedom experience fractionalization [62]. It may also be interesting to extend the framework of oblique TI phases to gauge theories involving non-Abelian bulk gauge groups, such as $SO(N)$. Such theories also lead to discrete emergent higher-form symmetries and would provide a natural avenue for further exploration of dyon condensation and oblique TI physics. It would also be interesting to consider the possibility of models with gapped, non-Abelian boundary states [63], which likely have a more intricate anomaly structure involving both ordinary global symmetries and the bulk one-form gauge symmetry.

*Note added:* After the completion of this manuscript, we became aware of the independent work, Ref. [65], which studies symmetry-protected topological phases of a multi-component generalization of the topological field theory we study.

## Acknowledgments

We especially thank Fiona Burnell and Ho Tat Lam for enlightening conversations during the development of this work. We also thank Arkya Chatterjee, Yu-An Chen, Tarun Grover, Chao-Ming Jian, Ethan Lake, John McGreevy, Salvatore Pace, Nathan Seiberg, T. Senthil, Steven Simon, Jun Ho Son, Nathanan Tantivasadakarn, and Xiao-Gang Wen for discussions and comments on the manuscript. HG is supported by the Gordon and Betty Moore Foundation EPiQS Initiative through Grant No. GBMF8684 at the Massachusetts Institute of Technology. RS was supported in part by the Natural Sciences and Engineering Research Council of Canada (NSERC) [funding reference number 6799-516762-2018]. This work was also supported in part by the US National Science Foundation through the NSF under grant No. DMR-1725401 at the University of Illinois (BM, EF).

# A    Review of generalized global symmetries

In this appendix we give an introduction to the concept of generalized global symmetries, introduced in Ref. [35]. For a recent review of generalized symmetries, see Ref. [45]. Here we will focus on $U(1)$ and $\mathbb{Z}_N$ gauge theories and how the generalized global symmetries manifest in the IR behavior of the phases of these theories. For a detailed discussion of phases of gauge theories and their characterization, see Refs. [66, 67].

## A.1    $U(1)$ Maxwell theory

We start by considering pure Maxwell theory in 3+1 spacetime dimensions,

$$S_{\text{Maxwell}} = \int d^4x \left[ -\frac{1}{4g^2} f^{\mu\nu} f_{\mu\nu} \right] . \tag{A.1}$$

Here $f_{\mu\nu} = \partial_\mu a_\nu - \partial_\nu a_\mu$, where $a_\mu$ is a compact $U(1)$ gauge field[14]. In the absence of monopoles, this theory describes a free photon, and therefore is in a *Coulomb phase*. In other words, lines of electric flux cost very little energy, and the Wilson loop generally decays as a perimeter law (or more slowly),

$$\text{charges are deconfined:} \quad \langle W_\Gamma \rangle = \left\langle e^{i \oint_\Gamma a} \right\rangle \sim e^{-\text{Length}(\Gamma)} , \tag{A.2}$$

where $\Gamma$ is a closed loop in spacetime. On introducing monopoles into the theory, their condensation leads to a *confining phase* of electric charges. Now electric flux is costly, and the Wilson loop has an area law,

$$\text{charges are confined:} \quad \langle W_\Gamma \rangle \sim e^{-\text{Area}(\Gamma)} . \tag{A.3}$$

This means that the Wilson line can be understood as an 'order parameter' for confinement (albeit a non-local one).

It is also useful to consider loops of magnetic charge, or 't Hooft loops, which we will denote $T_\Gamma$. 't Hooft loops fall into a general class of objects known as *disorder operators* [68] (for a general review on disorder operators, see Ref. [69] and references therein), which do not have a simple local form in terms of the gauge field operator. In the Coulomb phase,

---

[14]In this work, we use *compact* to mean that gauge fields are valued in e.g. $U(1)$ rather than $\mathbb{R}$, and as such have quantized fluxes. On the other hand, in some parts of the condensed matter literature, the word *compact* is used to denote theories for which the inclusion of monopole operators in the action is allowed. Because throughout this work our focus will be on lattice gauge theories, this latter sense will be implicitly understood to hold also.

where magnetic flux is also energetically inexpensive, 't Hooft loops also decay with the perimeter of the loop (or more slowly), while the confinement of monopoles leads to area law behavior. The natural context in which monopoles are confined is a superconductor, or *Higgs phase*: due to the Meissner effect, superconductors expel magnetic flux, meaning that monopoles can only appear in neutral pairs linked by a magnetic flux line (i.e. a vortex), which mediates a potential that is linear in the separation of the monopoles.

The existence of Wilson and 't Hooft loop operators, despite being non-local, suggests the possibility of a symmetry breaking scenario for confinement. Rather than being based on ordinary global symmetries, we now understand that such a notion can be based on *generalized global symmetries*. In the context of Maxwell theory, let us define the "electric" and "magnetic" two-form current operators,

$$J_E^{\mu\nu} = \frac{1}{g^2} f^{\mu\nu} \,, \qquad J_M^{\mu\nu} = \frac{1}{2\pi} \tilde{f}^{\mu\nu} = \frac{1}{4\pi} \varepsilon^{\mu\nu\lambda\sigma} f_{\lambda\sigma} \,. \tag{A.4}$$

In the absence of electric charges or monopoles, $\partial_\mu J_E^{\mu\nu} = 0$ by the equation of motion (Maxwell's equations), and $\partial_\mu J_M^{\mu\nu} = 0$ by the Bianchi identity. Thus, each of these currents is conserved, and we say that the theory thus possesses a $U(1)_E \times U(1)_M$ *one-form symmetry*, where we call $U(1)_E$ the electric one-form symmetry and $U(1)_M$ the magnetic one-form symmetry. A one-form symmetry is a symmetry that acts on loop operators. Here,

$$\begin{aligned} U(1)_E : &\qquad W_\Gamma \to e^{i\alpha_E} W_\Gamma \,, \\ U(1)_M : &\qquad T_\Gamma \to e^{i\alpha_M} T_\Gamma \,. \end{aligned} \tag{A.5}$$

In the case of $U(1)_E$, it is evident that this means that the one-form symmetry acts by shifting $a_\mu$ by a gauge connection, $\lambda_\mu$,

$$a_\mu \to a_\mu + \lambda_\mu \,, \qquad \oint_\Gamma \lambda = \alpha_E \,, \tag{A.6}$$

such that $J_E^{\mu\nu}$ and $J_M^{\mu\nu}$ are both invariant. $U(1)_M$ acts in the same way, but on the dual gauge field, $\tilde{a}_\mu$, coupling to monopoles. (One would obtain a gauge theory of this field by acting with electromagnetic duality.)

Therefore, $W_\Gamma$ and $T_\Gamma$ may be viewed as order parameters respectively for the electric and magnetic one-form symmetries. Indeed, we may view "perimeter law" behavior (or slower decay, such as "Coulomb law" in 3+1-D) for one of these loop operators as indicating breaking of the corresponding one-form symmetry, and it is also possible to generalize the notion of long-ranged order to these non-local order parameters. Furthermore, in the Coulomb phase, the gapless photon is sometimes regarded as a Goldstone mode for the broken $U(1)_E$ and

$U(1)_M$ one-form symmetries. The adaptation of the Landau symmetry breaking criterion to generalized global symmetries has been fleshed out in a number of recent works [35, 37, 38, 45, 70, 71].

We remark, however, that some concepts associated with spontaneous breaking of ordinary global symmetries do not have a clear analogue for generalized symmetries. For example, the only sharp definition of spontaneous symmetry breaking of ordinary global symmetries is the statement that upon coupling the *local* order parameter of the theory to a *local* symmetry breaking field, $h(x)$, the vacuum expectation value of the order parameter field in the thermodynamic limit remains nonzero as $h \to 0$. This, in turn, implies that the vacuum of the theory is not invariant under the global symmetry even in the absence of a symmetry breaking field. There is no precise analogue of this definition for generalized global symmetries due to the non-local nature of the observables. For the same reasons, there is no *precise* analogue of Goldstone's theorem for generalized global symmetries either, although it is possible to derive the presence of gapless degrees of freedom ("photons") in theories with broken generalized symmetries.

## A.2  $\mathbb{Z}_N$ gauge theories

Our primary interest in this work is in discrete, $\mathbb{Z}_N$ gauge theories, which can be obtained by condensing matter fields of charge $N$. If the matter is described by a current, $J^\mu$, current conservation in 3+1-D, $\partial_\mu J^\mu = 0$, implies we may re-express the matter current as the flux of a *two-form gauge field* (a Kalb-Ramond field) [72, 73], $b_{\mu\nu}$,

$$J^\mu = \frac{1}{2\pi} \frac{1}{2} \varepsilon^{\mu\nu\lambda\sigma} \partial_\nu b_{\lambda\sigma} \, . \tag{A.7}$$

We may therefore write the coupling to matter as a mutual Chern-Simons type-theory (in four dimensions),

$$S_{\mathrm{BF}} = N \int d^4x \, J^\mu a_\mu = \frac{N}{2\pi} \int b \wedge da \, , \tag{A.8}$$

where we have introduced the notation of forms, $b = \frac{1}{2} b_{\mu\nu} \, dx^\mu \wedge dx^\nu$, $a = a_\mu \, dx^\mu$, and $d$ is the exterior derivative. Throughout this work, we switch between component and form notation as needed to maximize clarity.

Because the matter is condensing, $b$ fluctuates wildly, hence acting as a Lagrange multiplier Higgsing all gauge configurations except those satisfying

$$\oint_\Gamma a = \frac{2\pi \, n}{N} \in \mathbb{Z}_N \, , \tag{A.9}$$

where $\Gamma$ is a closed loop. This is simply the statement that charge-$N$ superconductors have fractional vortices. Thus, the Lagrangian in Eq. (A.8) describes the spontaneous breaking of a $U(1)$ gauge theory down to a $\mathbb{Z}_N$ gauge theory, which has a $\mathbb{Z}_N$ electric one-form symmetry. In the absence of monopoles, the remaining uncondensed electric charges of magnitude between 1 and $N$ are deconfined, breaking the $\mathbb{Z}_N$ one-form symmetry. Examining the ground state degeneracy of this theory on closed manifolds, one finds that the theory has a topological ground state degeneracy and thus exhibits $\mathbb{Z}_N$ topological order. This topological order can be destroyed by condensing charge-1 monopoles, leading the uncondensed charges to confine and preserving the $\mathbb{Z}_N$ one-form symmetry.

In terms of one-form symmetries, the above discussion tells us that topological orders are characterized by the breaking of discrete one-form symmetries. Although in the example of ordinary $\mathbb{Z}_N$ gauge theory, the topologically ordered phase has no remaining electric one-form symmetry, in the main text, we study more ornate examples with $\Theta$-terms and oblique confinement. This idea provides an avenue for the construction of topologically ordered states having both topological order and a remaining discrete one-form global symmetry that can be the basis for a FTI phase.

## B  Canonical quantization of the TQFT

To supplement the path integral discussion of the effective TQFT in the main text in Section 5.3, we also examine this field theory from the perspective of canonical quantization. We work in Minkowski spacetime on the manifold $X = \mathbb{T}^3 \times \mathbb{R}$, where $\mathbb{R}$ represents the time direction and $\mathbb{T}^3$ is a spatial 3-torus of dimensions $R \times R \times R$. The action is

$$S = \frac{Nn}{2\pi} \int_X b \wedge da + \frac{Nnm}{4\pi} \int_X b \wedge b \tag{B.1}$$

$$= \frac{Nn}{4\pi} \int_X d^4x \; \varepsilon^{\mu\nu\lambda\sigma} (\partial_\mu a_\nu) b_{\lambda\sigma} + \frac{Nnm}{16\pi} \int_X d^4x \; \varepsilon^{\mu\nu\lambda\sigma} b_{\mu\nu} b_{\lambda\sigma} \tag{B.2}$$

$$= \frac{Nn}{4\pi} \int_X d^4x \; \left[ \varepsilon^{ijk} (\partial_t a_i) b_{jk} + \varepsilon^{ijk} a_0 \partial_i b_{jk} + 2 b_{0i} \varepsilon^{ijk} \partial_j a_k + m \, \varepsilon^{ijk} b_{0i} b_{jk} \right]. \tag{B.3}$$

We choose the axial gauge for both $a$ and $b$: $a_0 = b_{0i} = 0$. There are two Gauss law constraints, given by

$$\partial_i \left( \frac{1}{2} \varepsilon^{ijk} b_{jk} \right) = 0, \tag{B.4}$$

$$\varepsilon^{ijk} \partial_j a_k + m \left( \frac{1}{2} \varepsilon^{ijk} b_{jk} \right) = 0. \tag{B.5}$$

The solution for the first constraint is

$$b_{ij} = \frac{\bar{b}_{ij}(t)}{R^2} + \partial_i \xi_j - \partial_j \xi_i, \tag{B.6}$$

where $\xi_i$ is a periodic function on $\mathbb{T}^3$ and $\bar{b}_{ij}(t)$ depends only on time, $t$. Inserting this solution into the second constraint gives us

$$\varepsilon^{ijk} \partial_j a_k = -m \left( \frac{\frac{1}{2} \varepsilon^{ijk} \bar{b}_{jk}(t)}{R^2} + \varepsilon^{ijk} \partial_j \xi_k \right). \tag{B.7}$$

The solution for $a_i$ is then

$$a_i = \frac{\bar{a}_i(t)}{R} + \partial_i \Lambda + m \left( \frac{\frac{1}{2} \bar{b}_{ij}(t) x^j}{R^2} - \xi_i \right), \tag{B.8}$$

where $\Lambda$ is a periodic function on $\mathbb{T}^3$ and $\bar{a}_i(t)$ depends only on time. Substituting the solutions for $a_i$ and $b_{ij}$ into the action, we obtain

$$S = \frac{Nn}{2\pi} \int_X d^4x \left[ \frac{\frac{1}{2} \varepsilon^{ijk} \bar{b}_{jk}(t)}{R^2} \left( \frac{\dot{\bar{a}}_i(t)}{R} + \frac{m}{R^2} \frac{1}{2} \dot{\bar{b}}_{i\ell}(t) x^\ell \right) \right] \tag{B.9}$$

$$= \frac{Nn}{2\pi} \int_{\mathbb{R}} dt \frac{1}{2} \varepsilon^{ijk} \bar{b}_{jk}(t) \left( \dot{\bar{a}}_i(t) + \frac{m}{4} \sum_\ell \dot{\bar{b}}_{i\ell}(t) \right). \tag{B.10}$$

The action has reduced to that of an ordinary quantum mechanical system. The canonical commutation relations are

$$\left[ \bar{a}_i, \frac{1}{2} \varepsilon^{jk\ell} \bar{b}_{k\ell} \right] = i \frac{2\pi}{Nn} \delta_{ij}, \qquad [\bar{a}_i, \bar{a}_j] = [\bar{b}_{ij}, \bar{b}_{k\ell}] = 0. \tag{B.11}$$

By the Baker-Campbell-Hausdorff formula, the Wilson loops $\mathcal{W}_i = e^{i \bar{a}_i}$ and Wilson surfaces $\mathcal{U}_j = e^{i \varepsilon^{jk\ell} \bar{b}_{k\ell}/2}$ obey the algebra,

$$\mathcal{W}_j \mathcal{U}_k = e^{-2\pi i \delta_{jk}/Nn} \mathcal{U}_k \mathcal{W}_j,$$
$$\mathcal{W}_j \mathcal{W}_k = \mathcal{W}_k \mathcal{W}_j, \tag{B.12}$$
$$\mathcal{U}_j \mathcal{U}_k = \mathcal{U}_k \mathcal{U}_j,$$

which implies that $(\mathcal{W}_i)^{Nn}$ and $(\mathcal{U}_j)^{Nn}$ commute with all other operators of the theory, so we take $(\mathcal{W}_i)^{Nn} = (\mathcal{U}_j)^{Nn} = 1$. Furthermore, by Dirac quantization for $a_j$, the constraint in Eq. (B.5) implies that

$$\bar{b}_{jk} = -\frac{1}{m} \oint_{\Sigma_{jk}} (\partial_j a_k - \partial_k a_j) \, dx^j dx^k \in \frac{2\pi}{m} \mathbb{Z}. \tag{B.13}$$

Here, the repeated indices are not summed over, and $\Sigma_{jk}$ is a 2-torus, $\mathbb{T}^2 = S^1 \times S^1$, in the $j$ and $k$ directions. Thus, we also have $(\mathcal{U}_j)^m = 1$. Combining this condition with $(\mathcal{U}_j)^{Nn} = 1$, we find that $(\mathcal{U}_j)^L = 1$ where $L = \gcd(Nn, m)$. The result $(\mathcal{U}_j)^L = 1$ also has implications for $\mathcal{W}_j$ since $\bar{a}_i$ is canonically conjugate to $\frac{1}{2}\varepsilon^{ijk}\bar{b}_{jk}$. The commutation relation for $\bar{a}_i$ and $\frac{1}{2}\varepsilon^{ijk}\bar{b}_{jk}$ is consistent with $(\mathcal{U}_i)^L = 1$ only if $\bar{a}_i \sim \bar{a}_i + 2\pi L/Nn$. Therefore, the physical line operators are generated by $(\mathcal{W}_i)^{Nn/L}$ rather than $\mathcal{W}_i$.

Thus, we have line operators $(\mathcal{D}_i)^k = (\mathcal{W}_i)^{kNn/L}$ and surface operators $(\mathcal{U}_i)^k$, where $0 \le k < L$, which obey the algebra,

$$\mathcal{D}_j \mathcal{U}_k = e^{-2\pi i\, \delta_{jk}/L} \mathcal{U}_k \mathcal{D}_j,$$
$$\mathcal{D}_j \mathcal{D}_k = \mathcal{D}_k \mathcal{D}_j, \tag{B.14}$$
$$\mathcal{U}_j \mathcal{U}_k = \mathcal{U}_k \mathcal{U}_j.$$

Moreover, since the physical operators obey the same algebra as those of a topological $\mathbb{Z}_L$ gauge theory on a torus, the ground state degeneracy is the same, namely $L^3$. Hence, the results in this appendix agree with the operator content and correlation functions for the TQFT discussed in the path integral formalism in Section 5.3.

## C  Fermionic Cardy-Rabinovici model: Effective field theory

Typically, models with fundamental fermions must be placed on a spin manifold. In the fermionic Cardy-Rabinovici model, however, all magnetically neutral particles with odd electric charge are fermions while those with even electric charge are bosons. For condensed matter systems that obey this spin-charge relation, we can place the theory on a manifold with a spin$_c$ structure [74, 75], which is less restrictive since any oriented four-manifold admits a spin$_c$ structure (for a review, see Ref. [76]). When we do this, the electric charges should be coupled to a $U(1)$ spin$_c$ connection rather than a $U(1)$ gauge field. This suggests how to modify the effective field theory we obtained in Section 5.1 for the bosonic theory to capture the phases of the fermionic model.

For the fermionic CR model, the topological content of a $(n, m)$ oblique phase is described by

$$S = \frac{iNn}{2\pi} \int b \wedge da + \frac{iNnm}{4\pi} \int b \wedge b, \tag{C.1}$$

which is similar to the bosonic case discussed in Section 5, except that $a$ is now a $U(1)$ spin$_c$ connection instead of a $U(1)$ gauge field. A $U(1)$ spin$_c$ connection is locally the same as a

$U(1)$ gauge field but has a modified flux quantization,

$$\oint da = \oint \pi w_2 \mod 2\pi \mathbb{Z}, \tag{C.2}$$

where $w_2$ is the second Stiefel-Whitney class.

To better understand the physics of the effective field theory for the fermionic CR model, we show that a fermionic theory in which $(n, m)$ is condensed is dual to a bosonic theory with $(n, m + Nn)$ condensed. Our analysis is similar to that of Refs. [74, 77]. We dualize the spin$_c$ connection, $a$, by integrating over $f$ as a two-form field and introducing a $U(1)$ gauge field, $\tilde{a}$, that constrains $f = da$, where $a$ is a $U(1)$ spin$_c$ connection. This process gives the action,

$$S = \frac{iNn}{2\pi} \int b \wedge f + \frac{iNnm}{4\pi} \int b \wedge b + \frac{i}{2\pi} \int f \wedge d\tilde{a} + \frac{i}{4\pi} \int d\tilde{a} \wedge d\tilde{a}. \tag{C.3}$$

To understand the role of the last term, we note that

$$\frac{i}{4\pi} \int d\tilde{a} \wedge d\tilde{a} = -\int \pi w_2 \wedge \frac{d\tilde{a}}{2\pi} \mod 2\pi i. \tag{C.4}$$

Therefore, integrating over the $U(1)$ gauge field $\tilde{a}$ imposes both the local constraint $df = 0$ and the global constraint that $(f - \pi w_2)$ has cycles valued in $2\pi \mathbb{Z}$. Together, these constraints imply that $f = da$, where $a$ is a $U(1)$ spin$_c$ connection. Thus, we have properly dualized $a$.

If we proceed to integrate over $f$ and $b$, we obtain the action,

$$S_{\text{dual}} = \frac{i(m + Nn)}{4\pi Nn} \int d\tilde{a} \wedge d\tilde{a}. \tag{C.5}$$

However, we would have obtained the same result if we had instead acted with electromagnetic duality on the TQFT that describes the $(n, m + Nn)$ phase of the *bosonic* CR model,

$$S_{\text{bosonic}} = \frac{iNn}{2\pi} \int \widehat{b} \wedge d\widehat{a} + \frac{iNn(m + Nn)}{4\pi} \int \widehat{b} \wedge \widehat{b}, \tag{C.6}$$

where $\widehat{a}$ is a $U(1)$ one-form gauge field and $\widehat{b}$ is a $U(1)$ two-form gauge field. Thus, a fermionic phase in which $(n, m)$ is condensed has the same topological order as a bosonic phase in which $(n, m + Nn)$ is condensed.

Physically, this result makes sense because the $(n, m + Nn)$ phase of the bosonic CR model has $L$ deconfined dyons, where $L = \gcd(Nn, m + Nn) = \gcd(Nn, m)$, and the deconfined dyonic line operators,

$$D_\Gamma \left( q_e = \frac{kNn}{L}, q_m = \frac{k(m + Nn)}{L} \right) = (W_\Gamma)^{kNn/L} (T_\Gamma)^{k(m+Nn)/L}, \tag{C.7}$$

where $0 \leq k < L$, have self-statistics of

$$(-1)^{\frac{Nn(m+Nn)}{L^2}k^2} = (-1)^{\frac{Nnmk^2}{L^2}+\frac{Nnk}{L}}. \tag{C.8}$$

Thus, the topological data of the $(n, m+Nn)$ oblique phase of the bosonic CR model matches that of the $(n, m)$ phase of the fermionic CR model.

Finally, as mentioned in Section 8, we now show that the partition function is invariant under

$$\mathcal{T}: \quad (Nn, m) \mapsto (Nn - m, m). \tag{C.9}$$

To begin, we exploit the result that the $(n, m)$ phase of the fermionic CR model is dual to the $(n, m + Nn)$ phase of the bosonic CR model. As discussed in Section 8, the dualities of the TQFT for the bosonic theory are generated by $\mathcal{S}$ and $\mathcal{T}^2$, defined in Eq. (8.2). Thus, we can then apply these modular transformations to the $(n, m + Nn)$ state of the bosonic model to draw conclusions about the $(n, m)$ state of the fermionic theory. In particular, we find that

$$\mathcal{S}^2\mathcal{T}^{-2}\mathcal{S}: \quad (Nn, m + Nn) \mapsto (Nn - m, Nn). \tag{C.10}$$

By the duality argument presented earlier in this appendix, the state of the bosonic theory resulting from this transformation is dual to a fermionic theory on which we have acted with $\mathcal{T}$. Thus, we conclude that $\mathcal{T}$ preserves the topological order for the fermionic theory.

# D   Electromagnetic duality with boundaries

We consider how the boundary states in Section 7 transform under electromagnetic duality, adapting the duality calculation in Section 5.2.2 for when the theory is on a manifold with a boundary. We start with the electric boundary state studied in Section 7.1.1. The action is

$$S = \int_X \left( \frac{iNn}{2\pi} b \wedge da + \frac{iNnm}{4\pi} b \wedge b \right) + \int_{\partial X} \left( -\frac{iNnm}{4\pi} c \wedge dc + \frac{iNn}{2\pi} c \wedge da \right). \tag{D.1}$$

To study electromagnetic duality in the presence of a boundary, we perform manipulations similar to those in Refs. [77, 78]. We replace $da$ with $f$ in the bulk and introduce a two-form Lagrange multiplier, $\tilde{f}$, for the constraint, $f = da$. The new action is

$$S = \int_X \left( \frac{iNn}{2\pi} b \wedge f + \frac{iNnm}{4\pi} b \wedge b - \frac{i}{2\pi} \tilde{f} \wedge (f - da) \right) + \int_{\partial X} \left( -\frac{iNnm}{4\pi} c \wedge dc + \frac{iNn}{2\pi} c \wedge da \right). \tag{D.2}$$

The newly introduced two-form fields, $f$ and $\tilde{f}$, transform under the one-form gauge symmetry, Eq. (7.3), as

$$f \to f - m \, d\lambda, \qquad \tilde{f} \to \tilde{f} + Nn \, d\lambda. \tag{D.3}$$

The equation of motion for $f$ is $\tilde{f} = Nnb$, which results in the action,

$$S = \int_X \left( \frac{im}{4\pi Nn} \tilde{f} \wedge \tilde{f} + \frac{i}{2\pi} \tilde{f} \wedge da \right) + \int_{\partial X} \left( -\frac{iNnm}{4\pi} c \wedge dc + \frac{iNn}{2\pi} c \wedge da \right). \tag{D.4}$$

Integrating by parts for the second term in the bulk, we obtain

$$S = \int_X \left( \frac{im}{4\pi Nn} \tilde{f} \wedge \tilde{f} - \frac{i}{2\pi} d\tilde{f} \wedge a \right) + \int_{\partial X} \left( -\frac{iNnm}{4\pi} c \wedge dc + \frac{iNn}{2\pi} c \wedge da + \frac{i}{2\pi} a \wedge \tilde{f} \right). \tag{D.5}$$

Finally, we integrate over $a$ in the bulk, keeping the boundary value $a|_{\partial X}$ fixed, which gives $\tilde{f} = d\tilde{a}$, where $\tilde{a}$ is a $U(1)$ one-form gauge field. The action now becomes

$$S = \int_X \left( \frac{im}{4\pi Nn} d\tilde{a} \wedge d\tilde{a} \right) + \int_{\partial X} \left( -\frac{iNnm}{4\pi} c \wedge dc + \frac{iNn}{2\pi} c \wedge da + \frac{i}{2\pi} a \wedge d\tilde{a} \right). \tag{D.6}$$

The gauge field $\tilde{a}$ transforms under the one-form gauge symmetry as

$$\tilde{a} \to \tilde{a} + Nn \, \lambda. \tag{D.7}$$

We can then perform a Hubbard-Stratonovich transformation and introduce an auxiliary two-form field $\tilde{b}$ in the bulk that transforms under the one-form gauge symmetry by

$$\tilde{b} \to \tilde{b} - d\lambda. \tag{D.8}$$

The resulting action is

$$S_{\text{dual}} = \int_X \left( -\frac{im}{2\pi} \tilde{b} \wedge d\tilde{a} - \frac{iNnm}{4\pi} \tilde{b} \wedge \tilde{b} \right) + \int_{\partial X} \left( -\frac{iNnm}{4\pi} c \wedge dc + \frac{iNn}{2\pi} c \wedge da + \frac{i}{2\pi} a \wedge d\tilde{a} \right). \tag{D.9}$$

We refer to this action as the dual of Eq. (D.1).

We now check the topological order of the dual action. There are three types of line operators on the boundary that can be formed from $a$, $\tilde{a}$, and $c$,

$$\begin{aligned}
V_\Gamma &= \exp\left( i \oint_\Gamma a - im \oint_\Gamma c \right), \\
\widetilde{V}_\Gamma &= \exp\left( i \oint_\Gamma \tilde{a} + iNn \oint_\Gamma c \right), \\
\mathcal{D}_\Gamma &= \exp\left( i \frac{Nn}{L} \oint_\Gamma a + i \frac{m}{L} \oint_\Gamma \tilde{a} \right),
\end{aligned} \tag{D.10}$$

where $\Gamma \subset \partial X$. The equation of motion for $a$ makes $\widetilde{V}_\Gamma$ trivial and sets

$$\mathcal{D}_\Gamma = (V_\Gamma)^{Nn/L} . \tag{D.11}$$

The $\mathcal{D}_\Gamma$ operators represent the bulk quasiparticles, and since $(\mathcal{D}_\Gamma)^L = 1$, we observe that there are $|Nn|$ boundary anyons generated by $V_\Gamma$, and their correlation functions are

$$\left\langle (V_\Gamma)^k \, (V_{\Gamma'})^{k'} \right\rangle = \exp\left( \frac{2\pi i \, kk'm}{Nn} \, \varphi_{\text{link}}[\Gamma, \Gamma'] \right), \tag{D.12}$$

where $\varphi_{\text{link}}[\Gamma, \Gamma']$ is the linking number introduced in Eq. (7.11). Thus, the surface topological order is the same as for the original action, Eq. (D.1).

As previously noted in Section 7.1.1, the original boundary state in Eq. (D.1) is equivalent to setting the boundary condition $b|_{\partial X} = 0$. The fact that $\widetilde{V}_\Gamma$ is trivial means that this boundary state for the dual theory, Eq. (D.9), is equivalent to the boundary condition $\tilde{a}|_{\partial X} = 0$, which makes sense from examining the equations of motion we obtain in the duality calculation: $b = \tilde{f}/Nn = d\tilde{a}/Nn$.

We can perform the same manipulations with the magnetic boundary state, introduced in Section 7.1.2,

$$\widetilde{S} = \int_X \left( -\frac{iNn}{2\pi} db \wedge a + \frac{iNnm}{4\pi} b \wedge b \right) + \int_{\partial X} \left( \frac{iNnm}{4\pi} \tilde{c} \wedge d\tilde{c} + \frac{iNn}{2\pi} b \wedge (m\tilde{c} - d\phi) \right) \tag{D.13}$$

$$= \int_X \left( \frac{iNn}{2\pi} b \wedge da + \frac{iNnm}{4\pi} b \wedge b \right) + \int_{\partial X} \left( \frac{iNnm}{4\pi} \tilde{c} \wedge d\tilde{c} + \frac{iNn}{2\pi} b \wedge (m\tilde{c} - a - d\phi) \right). \tag{D.14}$$

Again, we replace $da$ with $f$ and integrate over $f$ as a two-form, introducing a Lagrange multiplier $\tilde{f}$ that constrains $f = da$. The action in Eq. (D.14) then becomes

$$\widetilde{S} = \int_X \left( \frac{iNn}{2\pi} b \wedge f + \frac{iNnm}{4\pi} b \wedge b - \frac{i}{2\pi} \tilde{f} \wedge (f - da) \right)$$
$$+ \int_{\partial X} \left( \frac{iNnm}{4\pi} \tilde{c} \wedge d\tilde{c} + \frac{iNnm}{2\pi} \tilde{c} \wedge b - \frac{iNn}{2\pi} b \wedge a - \frac{iNn}{2\pi} b \wedge d\phi \right). \tag{D.15}$$

The one-form gauge transformations for $f$ and $\tilde{f}$ are the same as before. Integrating over $f$, the resulting action is

$$\widetilde{S} = \int_X \left( \frac{im}{4\pi Nn} \tilde{f} \wedge \tilde{f} + \frac{i}{2\pi} \tilde{f} \wedge da \right) + \int_{\partial X} \left( \frac{iNnm}{4\pi} \tilde{c} \wedge d\tilde{c} + \frac{im}{2\pi} \tilde{c} \wedge \tilde{f} - \frac{i}{2\pi} \tilde{f} \wedge (a + d\phi) \right)$$
$$= \int_X \left( \frac{im}{4\pi Nn} \tilde{f} \wedge \tilde{f} - \frac{i}{2\pi} d\tilde{f} \wedge a \right) + \int_{\partial X} \left( \frac{iNnm}{4\pi} \tilde{c} \wedge d\tilde{c} + \frac{im}{2\pi} \tilde{c} \wedge \tilde{f} - \frac{i}{2\pi} \tilde{f} \wedge d\phi \right). \tag{D.16}$$

Integrating over $a$ in the bulk and $\phi$ on the boundary then gives that $\tilde{f} = d\tilde{a}$. The dual action is then

$$\widetilde{S}_{\text{dual}} = \int_X \left( \frac{im}{4\pi Nn} d\tilde{a} \wedge d\tilde{a} \right) + \int_{\partial X} \left( \frac{iNnm}{4\pi} \tilde{c} \wedge d\tilde{c} + \frac{im}{2\pi} \tilde{c} \wedge d\tilde{a} \right). \tag{D.17}$$

To make the result more transparent, we introduce an auxiliary two-form field $\tilde{b}$ to repackage the bulk term. The resulting action is

$$\widetilde{S}_{\text{dual}} = \int_X \left( -\frac{im}{2\pi} \tilde{b} \wedge d\tilde{a} - \frac{iNnm}{4\pi} \tilde{b} \wedge \tilde{b} \right) + \int_{\partial X} \left( \frac{iNnm}{4\pi} \tilde{c} \wedge d\tilde{c} + \frac{im}{2\pi} \tilde{c} \wedge d\tilde{a} \right). \tag{D.18}$$

This action is the dual of $\widetilde{S}$ in Eq. (D.14).

This result, Eq. (D.18), is the same as Eq. (D.1) except that the bulk gauge fields have been replaced by their duals and $(Nn, m) \to (m, -Nn)$. From the analysis of the theory in Eq. (D.1), we then know that this boundary theory has $|m|$ nontrivial genuine line operators, which are generated by

$$\widetilde{V}_\Gamma = \exp\left( i \oint_\Gamma \tilde{a} + iNn \oint_\Gamma \tilde{c} \right) = \exp\left( iNn \oint_\Gamma \tilde{c} + iNn \int_\Sigma b \right), \tag{D.19}$$

and have correlation functions of

$$\left\langle \left( \widetilde{V}_\Gamma \right)^k \left( \widetilde{V}_{\Gamma'} \right)^{k'} \right\rangle = \exp\left( -\frac{2\pi i\, kk'Nn}{m} \varphi_{\text{link}}[\Gamma, \Gamma'] \right), \tag{D.20}$$

so the topological order matches what we found for $\widetilde{S}$ in Section 7.1.2.

In conclusion, for both topological boundary states we analyzed, the number of anyons and their braiding data remains the same after performing electromagnetic duality. Duality preserves the topological order of the boundary state, maintaining the boundary condition of the original theory but changing its description. Thus, both of the two boundary states we examined can be expressed in terms of either the "electric" or "magnetic" fields. By examining Eqs. (D.9) and (D.18), it is clear that the electric boundary condition, Eq. (D.9), is not invariant under the global $\mathbb{Z}_m$ magnetic one-form symmetry, Eq. (5.14), while the magnetic boundary condition, Eq. (D.18), preserves this symmetry. Moreover, since the deconfined anyons for the magnetic boundary condition are generated by the operator, $\widetilde{V}_\Gamma$, defined in Eq. (D.19), which transforms under the magnetic one-form symmetry, Eq. (5.14), the topological order for this boundary state is realized by breaking the $\mathbb{Z}_m$ magnetic one-form symmetry completely.

# E  1+1-D Cardy-Rabinovici model

The Cardy-Rabinovici model can be dimensionally reduced to a $\mathbb{Z}_N$ spin model, or clock model, on a 2D Euclidean square lattice, which shares many qualitative features with the 4D lattice gauge theory [23, 24]. Here, we push this analogy further and demonstrate that the results in our work for the 4D Cardy-Rabinovici model have direct analogues in this 2D model. The partition function for the 2D Cardy-Rabinovici model is

$$Z = \int [d\varphi_I] \sum_{\{n_I, s_{\mu I}\}} e^{-S[n_I, \varphi_I, s_{\mu I}]} , \tag{E.1}$$

$$S = \frac{1}{2g^2} \sum_{r,R} (\Delta_\mu \varphi_I - 2\pi s_{\mu I})^2 - iN \sum_{r,R} n_I \, \varphi_I \tag{E.2}$$
$$+ \frac{iN\theta}{8\pi^2} \sum_{r,R} \varepsilon_{\mu\nu} \, \varepsilon_{IJ} \, (\Delta_\mu \varphi_I - 2\pi s_{\mu I}) \, (\Delta_\nu \varphi_J - 2\pi s_{\nu J}) ,$$

where, as in the main text, $r$ labels sites of the direct lattice, $R$ represents sites of the dual lattice, and $g^2$ is a coupling constant. Here, $\mu, \nu = \tau, x$ are Euclidean spacetime indices and $I, J = 1, 2$ are internal "flavor" indices. The lattice variables $\varphi_I \in \mathbb{R}$ and $n_I \in \mathbb{Z}$ live on sites, and the $s_{\mu I} \in \mathbb{Z}$ are on links. More specifically, variables with $I = 1$ ($I = 2$) are on sites or links of the direct (dual) lattice.

Physically, the analogue of an electric current, $n_I$, corresponds to spin wave excitations. The magnetic current is analogous to

$$m_I = -\varepsilon_{IJ} \, \varepsilon_{\mu\nu} \, \Delta_\mu s_{\nu J} , \tag{E.3}$$

which represents the vorticity of $\varphi_I$. When $\theta = 0$, the model in Eq. (E.1) describes two decoupled $\mathbb{Z}_N$ spin models. The last term Eq. (E.1) is the dimensional reduction of a $\Theta$-term, representing a four-spin interaction that couples the two $\mathbb{Z}_N$ models.

An advantage of this 2D model is that it is not necessary to introduce the non-local interaction, $K(x)$, in Eq. (3.1). Additionally, a more precise analogue of the condensation criterion for the 4D model, Eq. (3.10), may be constructed and is supported by a renormalization group analysis [23]. Indeed, the criterion for an excitation with quantum numbers $(n_I, m_I) = (n, m)$ to condense is

$$\frac{2\pi}{Ng^2} m^2 + \frac{Ng^2}{2\pi} \left( n + \frac{\theta}{2\pi} m \right)^2 < \frac{4}{N} . \tag{E.4}$$

This inequality matches Eq. (3.10) with $C = 4$. We thus see that the 2D spin model has a phase diagram that is similar to the 4D lattice gauge theory. In particular, the phase with

$(n_I, m_I) = (n, m)$ condensed is stable in the limit $g^2 \to \infty$, $\theta/2\pi = -n/m$. Following the analysis for the 4D model in Section 5.1, we can then examine this limit to determine the effective field theory.

In the strong-coupling limit, $g^2 \to \infty$, the action becomes

$$S = \sum_{r,R} \left[ \frac{iN\theta}{8\pi^2} \, \varepsilon_{\mu\nu} \, \varepsilon_{IJ} \left( \Delta_\mu \varphi_I \right) \left( \Delta_\nu \varphi_J \right) - iN \left( n_I + \frac{\theta}{2\pi} m_I \right) \varphi_I + \frac{iN\theta}{8\pi^2} \left( 2\pi \right)^2 \varepsilon_{\mu\nu} \, \varepsilon_{IJ} \, s_{\mu I} \, s_{\nu J} \right]. \tag{E.5}$$

Plugging in $\theta/2\pi = -n/m$ and integrating out $\varphi_I$ gives the local constraint,

$$n \, m_I = m \, n_I \,, \tag{E.6}$$

which is the analogue of Eq. (5.2). The solution is that $n_I = n \, j_I$ and $m_I = m \, j_I$ where $j_I \in \mathbb{Z}$. One then arrives at an effective theory consisting of $s_{\mu I}$ along with the constraint in Eq. (E.6),

$$Z_{(n,m)} = \sum_{\{s_{\mu I}, j_I\}} \delta\left( m_I[s_{\mu J}] - m j_I \right) \exp\left( -(2\pi)^2 \sum_{r,R} \frac{iNn}{4\pi m} \, \varepsilon_{\mu\nu} \, \varepsilon_{IJ} \, s_{\mu I} s_{\nu J} \right) \,, \tag{E.7}$$

where $\delta(x - y)$ is a Kronecker delta function defined on integer-valued lattice fields. As in Section 5.1, we "integrate in" the constraint, $m_I[s] = m j_I$, by introducing integer-valued variables, $\alpha_I \in \mathbb{Z}$,

$$S = \sum_{r,R} \left( \frac{2\pi i}{m} m_I[s] \, \alpha_I + (2\pi)^2 \frac{iNn}{4\pi m} \, \varepsilon_{\mu\nu} \, \varepsilon_{IJ} \, s_{\mu I} s_{\nu J} \right) \tag{E.8}$$

$$= \sum_{r,R} \left( \frac{im}{2\pi} \, \varepsilon_{\mu\nu} \, \varepsilon_{IJ} \, \frac{2\pi \alpha_J}{m} \frac{2\pi \Delta_\mu s_{\nu I}}{m} + \frac{iNnm}{4\pi} \, \varepsilon_{IJ} \, \varepsilon_{\mu\nu} \, \frac{2\pi s_{\mu I}}{m} \frac{2\pi s_{\nu J}}{m} \right) \,. \tag{E.9}$$

This action has an emergent zero-form gauge symmetry,

$$s_{\mu I} \to s_{\mu I} + \Delta_\mu \eta_I + m \, \mathcal{N}_{\mu I} \,, \qquad \alpha_I \to \alpha_I - Nn \, \eta_I \,, \tag{E.10}$$

where $\eta_I, \mathcal{N}_{\mu I} \in \mathbb{Z}$.

Applying the same reasoning as in Section 5.1, we determine the continuum limit of Eq. (E.9), which is a TQFT that can be written in terms of $2\pi$-periodic scalar fields, $\phi_I$, and $U(1)$ one-form gauge fields, $(a_I)_\mu$. The correspondence with the lattice variables is

$$2\pi \frac{\varepsilon_{IJ} \, \alpha_J}{m} \to \phi_I \,, \qquad 2\pi \frac{s_{\mu I}}{m} \to (a_I)_\mu \,, \tag{E.11}$$

and the action for the effective field theory is given by

$$S = \int \left( \frac{im}{2\pi} \phi_I \wedge da_I + \frac{iNnm}{4\pi} \varepsilon_{IJ} \, a_I \wedge a_J \right). \tag{E.12}$$

This TQFT has been discussed in detail in Refs. [35, 60] and represents a continuum formulation of a Dijkgraaf-Witten theory [79]. Here, we will briefly review the global symmetries, gauge invariant operators, and correlation functions. The operators of the TQFT may be matched easily with operators of the lattice model using the methods of Section 4.

The action in Eq. (E.12) has a global emergent $\mathbb{Z}_m \times \mathbb{Z}_m$ zero-form symmetry,

$$\phi_I \to \phi_I + \frac{2\pi}{m}. \tag{E.13}$$

Additionally, when the 2D spacetime has no boundary, the action, Eq. (E.12), is invariant under the gauge symmetry,

$$a_I \to a_I + d\xi_I, \qquad \phi_I \to \phi_I - \varepsilon_{IJ} \, Nn \, \xi_J, \tag{E.14}$$

where $\xi_I$ is a $2\pi$-periodic scalar. This gauge symmetry is the continuum analogue of Eq. (E.10). The local operators in the bulk are generated by

$$D_{\mathcal{P}}^{(I)} = \exp \left( i\frac{m}{L}\phi_I(\mathcal{P}) - i\frac{m}{L}\phi_I(\mathcal{P}') + i\frac{Nnm}{L} \int_\Gamma \varepsilon_{IJ} \, a_J \right), \tag{E.15}$$

where $\mathcal{P}$ and $\mathcal{P}'$ are points and $\Gamma$ is a curve from $\mathcal{P}'$ to $\mathcal{P}$. Local operators in this theory must be attached to a line operator to ensure gauge invariance, but the operators, $D_{\mathcal{P}}^{(I)}$, are such that the attached line operator is undetectable. We can also take $\mathcal{P}'$ to be at infinity (or some other particular point), so $D_{\mathcal{P}}^{(I)}$ is in fact a genuine local operator. These genuine local operators are analogous to the genuine loop operators, $\mathcal{D}_\Gamma$, for the 4D TQFT we discuss in Section 5.3. The number of genuine local operators is $L^2$, where $L = \gcd(Nn, m)$ is defined as in the main text. This counting of local operators also makes sense from the perspective of the lattice model. In a phase where $(n_I, m_I) = (n, m)$ is condensed, there are $L$ independent local operators for each flavor, labeled by $I = 1, 2$. Because we can construct arbitrary products of local operators for each flavor, in total there are $L^2$ local operators. Physically, the $D_{\mathcal{P}}^{(I)}$ operators are order parameters for the global symmetry $\mathbb{Z}_m \times \mathbb{Z}_m$, which is spontaneously broken to the subgroup $\mathbb{Z}_{m/L} \times \mathbb{Z}_{m/L}$. In the special case $L = 1$, there are no nontrivial operators, so the bulk theory is trivial and preserves the full $\mathbb{Z}_m \times \mathbb{Z}_m$ symmetry.

In addition, there are $L^2$ bulk loop operators, generated by

$$U_\Gamma^{(I)} = \exp \left( i \oint_\Gamma a_I \right), \tag{E.16}$$

where $\Gamma$ is a closed loop. These operators are analogous to the closed surface operators, $\mathcal{U}_\Sigma$, of the 4D TQFT. Surrounding $V_\mathcal{P}^{(I)}$ by $U_\Gamma^{(I)}$ leads to a phase, $e^{2\pi i/L}$, which resembles the mutual statistics between loop and surface operators for the 4D TQFT, Eq. (5.26). In this case, however, the interpretation is that the operators, $U_\Gamma^{(I)}$, represent domain walls, so moving an order parameter operator, $D_\mathcal{P}^{(I)}$, across a domain wall changes its value by a phase factor.

We can also construct a bulk response analogous to the generalized magnetoelectric effect discussed in Section 6. If we introduce background fields, $A_1$ and $A_2$, to the action, Eq. (E.12), for the $\mathbb{Z}_m \times \mathbb{Z}_m$ global symmetry, we obtain

$$S[A_1, A_2] = \int \left( \frac{im}{2\pi} a_I \wedge (d\phi_I + A_I) + \frac{iNnm}{4\pi} \varepsilon_{IJ} a_I \wedge a_J + \frac{im}{2\pi} A_I \wedge d\beta_I \right). \tag{E.17}$$

Here, the $\beta_I$ are a $2\pi$-periodic scalar fields that Higgs the background $U(1)$ gauge fields, $A_I$, to make them probes for the $\mathbb{Z}_m \times \mathbb{Z}_m$ symmetry. Integrating out $a_I$ leads to

$$S_{\text{eff}}[A_1, A_2] = \frac{im}{2\pi Nn} \int (d\phi_1 + A_1) \wedge (d\phi_2 + A_2) + \frac{im}{2\pi} \int A_I \wedge d\beta_I. \tag{E.18}$$

The $\beta_I$ fields constrain the $A_I$ to be $\mathbb{Z}_m$ gauge fields, and eliminating the $\phi_I$ by fixing to the unitary gauge gives the response,

$$S_{\text{response}}[A_1, A_2] = \frac{im}{2\pi Nn} \int A_1 \wedge A_2. \tag{E.19}$$

This result is directly analogous to the generalized magnetoelectric effect in Eq. (6.18).

Next, continuing in analogy with the 4D case, we examine possible boundary states. When the theory is defined on a manifold, $Y$, with a boundary, the action in Eq. (E.12) is no longer gauge invariant but instead changes by a boundary term. As in the 4D case, we can construct two distinct boundary states. One possible state has action,

$$S_1 = \int_Y \left( -\frac{im}{2\pi} d\phi_I \wedge a_I + \frac{iNnm}{4\pi} \varepsilon_{IJ} a_I \wedge a_J \right) + \int_{\partial Y} \left( -\frac{im}{2\pi} \phi_I \, d\chi_I - \frac{iNnm}{4\pi} \varepsilon_{IJ} \chi_I \, d\chi_J \right), \tag{E.20}$$

where $\chi_I$ is a $2\pi$-periodic scalar field defined on $\partial Y$ that transforms under the zero-form gauge symmetry, Eq. (E.14), by

$$\chi_I \rightarrow \chi_I - \xi_I. \tag{E.21}$$

There are $m^2$ genuine local operators on the boundary, generated by

$$\widetilde{V}_\mathcal{P}^{(I)} = \exp\left[ i\,\phi_I(\mathcal{P}) - i\,\varepsilon_{IJ}\,Nn\,\chi_J(\mathcal{P}) \right], \tag{E.22}$$

where $\mathcal{P}$ is a boundary point. Suppose we perform a Wick rotation on Eq. (E.20) and canonically quantize the theory on the spacetime $[0, L_0] \times \mathbb{R}$ where $L_0$ is a length and $\mathbb{R}$ represents time. Then, the operators, $\widetilde{V}_{\mathcal{P}}^{(I)}$, obey the algebra,

$$\widetilde{V}_{\mathcal{P}}^{(1)} \widetilde{V}_{\mathcal{P}}^{(2)} = e^{-2\pi i \, Nn/m} \, \widetilde{V}_{\mathcal{P}}^{(2)} \, \widetilde{V}_{\mathcal{P}}^{(1)}, \tag{E.23}$$

$$\widetilde{V}_0^{(I)} \widetilde{V}_{L_0}^{(J)} = \widetilde{V}_{L_0}^{(J)} \, \widetilde{V}_0^{(I)}, \tag{E.24}$$

where $\mathcal{P} \in \{0, L_0\}$ is a boundary point. We then see that this boundary state, Eq. (E.20), is the analogue of the 2+1-D magnetic boundary state discussed in Section 7.1.2. Eq. (E.23) shows that the boundary operators, $\widetilde{V}_{\mathcal{P}}^{(I)}$, are parafermion operators, which are emergent excitations that generalize Majorana fermions [80, 81]. We also note that although the operators at the ends of the space commute with one another, they are not truly independent. For example, we have

$$\widetilde{V}_0^{(2)} = \widetilde{V}_{L_0}^{(2)} \exp \left( -iNn \int_0^{L_0} a_1 \right), \tag{E.25}$$

implying that local operators at each end are connected by a string operator.

An alternate boundary state, is represented by the action,

$$S_2 = \int_Y \left( \frac{im}{2\pi} \phi_I \wedge da_I + \frac{iNnm}{4\pi} \varepsilon_{IJ} \, a_I \wedge a_J \right) + \int_{\partial Y} \left( \frac{iNnm}{2\pi} \varepsilon_{IJ} \widetilde{\chi}_I \, a_J + \frac{iNnm}{4\pi} \varepsilon_{IJ} \, \widetilde{\chi}_I \, d\widetilde{\chi}_J \right), \tag{E.26}$$

where $\widetilde{\chi}_I$ is a $2\pi$-periodic scalar field defined on $\partial Y$ that transforms under the zero-form gauge symmetry, Eq. (E.14), by

$$\widetilde{\chi}_I \to \widetilde{\chi}_I - \xi_I. \tag{E.27}$$

This boundary state has $(Nn)^2$ local operators, which are generated by

$$V_{\mathcal{P}}^{(I)} = \exp \left( im \, \widetilde{\chi}_I(\mathcal{P}) + im \int_\Gamma a_I - im \, \widetilde{\chi}_I(\mathcal{P}') \right), \tag{E.28}$$

where $\Gamma$ is a curve with endpoints at $\mathcal{P}$ and $\mathcal{P}'$. Like the bulk local operators in Eq. (E.15), these operators must be attached to a line operator to ensure gauge invariance, but the dependence on the line operator is trivial because of the factor of $m$. These operators obey the algebra,

$$V_{\mathcal{P}}^{(1)} V_{\mathcal{P}}^{(2)} = e^{2\pi i \, m/Nn} \, V_{\mathcal{P}}^{(2)} V_{\mathcal{P}}^{(1)}, \tag{E.29}$$

so we again find parafermion modes at the boundaries. Thus, the state described by Eq. (E.26) is analogous to the 2+1-D electric boundary state in Section 7.1.1.

To summarize, in a given phase where $(n_I, m_I) = (n, m)$ condenses, the effective field theory is captured by a TQFT, which leads to two distinct boundary states, and each of

these states contains parafermion operators. Thus, for the 2D Cardy-Rabinovici model, we find much of the same physics that we uncovered for the 4D Cardy-Rabinovici model in the main text.

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
