# Peer review of "Theory of oblique topological insulators"

_SciPost Physics_

## Round 2 · Referee Report · Anonymous (Referee 1) · 2022-8-2

Report

The manuscript discusses the oblique confinement phase and boundary states of the Cardy-Robinovici model and its continuum field theory, as well as the one-form symmetry in the theory. The manuscript also discusses an interesting duality in the model and in the continuum two-form gauge theory.

Here are some comments and questions:

  • p20 the authors says a Z_L subgroup one-form symmetry is broken by deconfined dyon loops, but it is a quotient rather than a subgroup (the unbroken symmetry is a subgroup)

  • p33 the author discusses the response by gauge-fixing a background gauge field to eliminate a dynamical gauge field. Can the author clarify why this is allowed (e.g. in ordinary Z_2 gauge theory coupled to background Z_2 two-form gauge field for the electric one-form symmetry, one cannot gauge fix the background two-form gauge field to eliminate the dynamical Z_2 gauge field and say the theory is trivial)

  • in the two-form gauge theory (6.11), the parameter m has equivalence relation m ~ m+2Nn. This is because b has holonomy 2pi/(Nn) * integer (note that it is not affected by the coupling to background B). But the response (6.18) is not invariant under such an identification of m. Can the author clarify this?

  • validity: -
  • significance: -
  • originality: -
  • clarity: -
  • formatting: -
  • grammar: -

Author:  Benjamin Moy  on 2022-09-14  [id 2812]

(in reply to Report 1 on 2022-08-02)

We thank the referee for useful comments and questions.

Referee's comment:

p20 the authors says a Z_L subgroup one-form symmetry is broken by deconfined dyon loops, but it is a quotient rather than a subgroup (the unbroken symmetry is a subgroup)

Our response:

We thank the referee for catching this misprint. We have corrected the manuscript accordingly.

Referee's question:

p33 the author discusses the response by gauge-fixing a background gauge field to eliminate a dynamical gauge field. Can the author clarify why this is allowed (e.g. in ordinary Z_2 gauge theory coupled to background Z_2 two-form gauge field for the electric one-form symmetry, one cannot gauge fix the background two-form gauge field to eliminate the dynamical Z_2 gauge field and say the theory is trivial)

Our response:

Eliminating the dynamical gauge field by gauge fixing the background field is only allowed if the bulk is topologically trivial (i.e. if $L=\gcd(Nn,m)=1$). We state this more explicitly in the updated version of the manuscript in the paragraph just below Eq. (6.18).

Referee's question:

in the two-form gauge theory (6.11), the parameter m has equivalence relation m ~ m+2Nn. This is because b has holonomy 2pi/(Nn) * integer (note that it is not affected by the coupling to background B). But the response (6.18) is not invariant under such an identification of m. Can the author clarify this?

Our response:

The periodicity in $m$ is equivalent to the periodicity of the $\Theta$-angle. As the referee correctly points out, this periodicity is no longer present if there is a background field, $B_{\mu\nu}$, as in Eq. (6.17). This lack of periodicity is not problematic. In fact, it has nontrivial physical implications. Namely, as discussed in Refs. [33] and [37], this is the avatar of a mixed anomaly between $\mathbf{CP}$ symmetry and the global one-form symmetry.

Anonymous on 2022-09-19  [id 2827]

(in reply to Benjamin Moy on 2022-09-14 [id 2812])

The periodicity of m is more manifest using discrete notation: consider Z_{Nn} two-cocyle b, the action (6.11) is
2\pi m/(2Nn) int P(b) + 2pi/N int b \cup B
where P is the Pontryagin square, and B is a background Z_N two-cocycle.
b has holonomy 0,1,... Nn-1 mod Nn, and B has holonomy 0,...N-1 mod N.
Then shifting m by 2Nn changes the action by 2pi (2Nn)/(2Nn) * integer = 2pi * integer, so m has periodicity 2Nn, even in the presence of background B. Does it make sense?

Author:  Benjamin Moy  on 2022-09-28  [id 2858]

(in reply to Anonymous Comment on 2022-09-19 [id 2827])

We thank the referee for raising this subtle question. Physically, taking $m\to m+2Nn$ means that we are changing magnetic charge of the dyon condensing in the UV from $m$ to $m+2Nn$. In the IR, although the bulk topological order is invariant under $m\to m+2Nn$, any aspects of the theory that depend on the UV magnetic charge will be different. In particular, the global magnetic one-form symmetry will not be the same. This becomes especially acute in the presence of a boundary: The "magnetic boundary condition" discussed in Section 7.1.2 is not invariant under $m\to m+2Nn$ because this state preserves the global magnetic one-form symmetry (but not the electric one-form symmetry). When the theory is on a manifold without a boundary, the response, Eq. (6.18), is the only way to distinguish a state with $(n,m)$ condensed from a phase with $(n,m+2Nn)$ condensed. (Recall that electric charge is given in units of $N$ in our notation.)

Our claim, then, is that the conclusion that $m\to m+2Nn$ leaves the theory invariant is naive. At the level of the action, one way this can be understood is by noticing that changing $m$ also necessarily modifies the one-form gauge symmetry. For example, in the Higgs phase, where we have condensed charge-$N$ electric particles, the IR theory is described by BF theory at level $N$,
$$S_\mathrm{BF}=\frac{iN}{2\pi}\int b\wedge da,$$
which has the one-form gauge symmetry $b\to b+d\lambda$, $a\to a$ where $\lambda$ is a $U(1)$ gauge field. Suppose we now instead consider an oblique phase with $(1,2N)$ condensed,
$$S=\frac{iN}{2\pi}\int b\wedge da+\frac{i(2N)N}{4\pi}\int b\wedge b.$$
Because of the second term, the theory is no longer invariant under the original one-form gauge symmetry. Instead, the one-form gauge symmetry must be modified to $b\to b+d\lambda$, $a\to a-2N\lambda$. We then have a different theory, which nonetheless has the same bulk topological order.

---

## Round 2 · Referee Report · Anonymous (Referee 2) · 2022-8-24

Weaknesses

  1. The paper is hard to follow.
  2. The conditions for stability of the topological phases is not stated in a clear or convincing manner.

Report

I think I broadly understood the paper, which provides a connection between axion field theories and certain lattice models. While this isn't the first work to notice the connection [32,59], the authors seem to do a more complete job.

Sadly, I still found this paper difficult to follow. Here are some questions.

  1. Under what conditions do the confined cardy rabinovici models constitute stable phases of matter? The authors will probably say that the phase is 1-form symmetry protected phase, where the 1-form symmetry is Eq 2.5?

  2. However, the authors also say that this 1-from symmetry is emergent. What precisely do they mean by this? Are they saying that if I break the 1-form symmetry in the UV, then it is re-instated in the IR? (That's the sort of statement I usually associated with words "emergent 1-form symmetry"). Wouldn't that imply that even the confined FTIs are stable to arbitrary (suff small) local perturbations?

  3. pg 8. "The charge-N electric charges are now confined and are projected out of the spectrum." How can they be confined when there's a charge-N Higgs field in the background? Wouldn't they be screened rather than confined?

  4. Can the authors make a clearer statement about how their results supersede those of [32]? It seems like there's a lot of overlap, except perhaps [32] covers only the case where $\theta/2\pi$ is an integer?

Requested changes

  1. Give a self-contained summary of the necessary and sufficient conditions for the stability of the FTIs discussed in this work.
  2. To head off potential confusion, the authors should mention that these FTIs are not traditional SPTs protected by a global symmetry. Most people associate words topological insulator with a regular 0-form SPT.
  3. Address points 3,4 above.

  • validity: good
  • significance: ok
  • originality: good
  • clarity: ok
  • formatting: good
  • grammar: good

Author:  Benjamin Moy  on 2022-09-14  [id 2813]

(in reply to Report 2 on 2022-08-24)

We thank the referee for comments and questions. We provide detailed responses to the questions below:

Referee's question:

Under what conditions do the *confined* cardy rabinovici models constitute stable phases of matter? The authors will probably say that the phase is 1-form symmetry protected phase, where the 1-form symmetry is Eq 2.5?

Our response:

The oblique confining phases of the Cardy-Rabinovici model are locally (or IR) stable, gapped phases of matter. Each such phase has distinct universal features: quasiparticle/string excitation spectra, global symmetries and response, ground state degeneracy, etc. Passing from one oblique confining state to another requires a (first or second order) quantum phase transition.

Nevertheless, one may naturally worry about the energetic stability of different oblique TI phases from the UV point of view of the Cardy-Rabinovici lattice gauge theory. While it is not possible to rigorously derive the energetic favorability of any given phase—a common limitation in studying correlated phases in both quantum and classical statistical mechanics—the original work of Cardy and Rabinovici (Ref. [23]) presented a heuristic free energy argument to determine the stability of each phase given some combination of $\Theta$ and $g^2$, the Maxwell coupling. We review this argument in Section 3.2 of the manuscript.

Interestingly, for an analogous model in $(1+1)$-D, it is in fact possible to derive a precise bound determining the energetic stability of each phase, and the physics is analogous. (See Appendix E and Ref. [23].)

Referee's question:

However, the authors also say that this 1-from symmetry is emergent. What precisely do they mean by this? Are they saying that if I break the 1-form symmetry in the UV, then it is re-instated in the IR? (That's the sort of statement I usually associated with words "emergent 1-form symmetry"). Wouldn't that imply that even the confined FTIs are stable to arbitrary (suff small) local perturbations?

Our response:

First, we emphasize that while there is an exact one-form symmetry, $\mathbb{Z}_N$, at the level of the UV lattice gauge theory (LGT), each low energy oblique TI phase possesses a different, emergent one-form symmetry, $\mathbb{Z}_{Nn}$. This symmetry is emergent in the sense that it appears as one goes to low energies (say along the renormalization group flow). This is not an especially exotic feature. In the majority of instances where global one-form symmetries appear and have non-trivial consequences, they are not only emergent but “spontaneously” broken (in the sense that they act non-trivially on topological operators at the IR fixed point).

The particular one-form symmetry one finds to emerge at low energies is determined by the low energy charged fluctuations, which may be bound states of the microscopic charges with one another or with monopoles. For example, we may regard the $\mathbb{Z}_N$ one-form symmetry of the LGT (or even of the simpler example of the toric code) as emergent if we imagine the theory as arising from a higher energy theory of more elementary, charge-1 matter.

In the case of an oblique TI phase, the ultimate low energy one-form symmetry is associated with the particular dyon that is condensing. More explicitly, because the only electric charges in the LGT are multiples of $N$, if a dyon of electric charge $Nn$ and magnetic charge $m$ condenses, then dyon fluctuations will dominate at low energies, and a different $\mathbb{Z}_{Nn}$ electric one-form symmetry emerges.

Referee's question:

pg 8. "The charge-N electric charges are now confined and are projected out of the spectrum." How can they be confined when there's a charge-N Higgs field in the background? Wouldn't they be screened rather than confined?

Our response:

In the UV, the lattice gauge theory has charge-$N$ electric matter and charge-1 monopoles. For small enough Maxwell coupling, $g^2$, loops of charge-$N$ electric matter become extremely energetically cheap. The purely electric loops therefore condense, resulting in a Higgs phase. In this case, the electric charges are indeed screened.

The physics is much richer at strong coupling. As $g^2$ is increased, composite loops of charges and/or monopoles, i.e. dyons, can become cheaper than the charge-$N$ electric loops, leading to the complex phase diagram of Figure 4. Just as monopoles are confined in an ordinary superconductor, in the case where dyon loops proliferate, all purely electric charges will be confined. The only effect of the presence charge-$N$ electric matter in the UV is that it just constrains the dyons that can possibly condense to have an electric charge that is a multiple of $N$.

Referee's question:

Can the authors make a clearer statement about how their results supersede those of [32]? It seems like there's a lot of overlap, except perhaps [32] covers only the case where $\theta/2\pi$ is an integer?

Our response:

Although we learned a lot and took much inspiration from Ref. [32], our work makes a great deal of important progress beyond what was done in that work. At a high level, we introduce oblique confinement as a novel mechanism for obtaining fractional TI phases equipped with emergent electric and magnetic one-form symmetries (that in turn have a mixed anomaly). While these symmetries are partially broken in the IR “spontaneously,” an interesting feature of these types of FTI phases is that they have boundary physics uniquely characterized by the remaining unbroken one-form symmetry. This is very exotic: The vast majority of cases in which an emergent higher-form symmetry has physical consequences involve cases where the symmetry is completely broken. We provide a comprehensive understanding of these types of phases, which is an important step toward determining the general types of higher-form symmetry protected/enriched topological phases in three dimensions.

More specifically:

  1. We provide a detailed accounting for each oblique confining phase of its global one-form symmetries and universal response, both at the level of the lattice gauge theory and that of the continuum TQFT for a particular phase. This is absent in Ref. [32] since the notion of higher-form global symmetries did not exist in the literature at the time.
  2. We also examine the response to probes for the one-form symmetry of the lattice model, developing a notion of a generalized magnetoelectric effect.
  3. Although similar TQFTs have been discussed in the literature, our derivation starting from the Cardy-Rabinovici lattice gauge theory is new and provides a physically transparent origin for the two-form gauge field of the TQFT. This approach allows access to the entire phase diagram and opens up possibilities to explore e.g. quantum phase transitions between different oblique TI phases. This is not possible if one adopts Walker-Wang models as a starting point, as these constructions are exactly solvable models of each particular phase.
  4. Ref. [32] considers only two of the oblique confining phases—the $(-1,1)$ and $(-1,2)$ phases in our notation. (Note that our notation is distinct from that of Ref. [32].)
  5. We give a more precise and complete discussion of the boundary states, which are novel types of fractional quantum Hall states not realizable in $(2+1)$-D alone, and we develop a new class of boundary states not previously considered.
  6. We discuss electromagnetic duality for the TQFT and clarify its distinction from the Cardy-Rabinovici duality introduced in Ref. [24].

Anonymous on 2022-09-20  [id 2830]

(in reply to Benjamin Moy on 2022-09-14 [id 2813])
Category:
objection

"Passing from one oblique confining state to another requires a (first or second order) quantum phase transition." Are you sure? The question was specifically about the confining phases (which don't have bulk topological order). In the confined case can't you add a higgs field with smaller charge to the theory, detune the axion term, and then remove the higgs field to return to a trivial confined phase. (see discussion on pg 14 of [32])?

Author:  Benjamin Moy  on 2022-09-28  [id 2857]

(in reply to Anonymous Comment on 2022-09-20 [id 2830])

In our notation, the authors of Ref. [32] argue on p. 14 that the $(-1,1)$ phases for *different* values of $N\in 2\mathbb{Z}$ are adiabatically connected. The $(-1,1)$ phase is the one that exists at $\theta\equiv \Theta/N=2\pi$ and $g^2\to\infty$. In our view, since the Cardy-Rabinovici theory is a $\mathbb{Z}_N$ gauge theory, introducing a Higgs field with a charge smaller than $N$ changes the theory. It is not natural to consider transitions between phases of different theories. Moreover, there is no way to add or remove Higgs fields adiabatically.

Nonetheless, if we fix $N$, we may still consider whether there must be a phase transition between the $(0,1)$ confining phase (at $g^2\to\infty, \theta=0$) and the $(-1,1)$ oblique confining phase (at $g^2\to\infty,\theta=2\pi$). Although both phases have no topological order in the bulk, they have distinct global electric one-form symmetries, so they will have different higher-form magnetoelectric effects and different boundary states (at least for the "electric" boundary condition discussed in Section 7.1.1). These phases are distinct as one-form symmetry-protected SPTs. One way to see this is that there is a mixed 't Hooft anomaly between time-reversal symmetry and the electric one-form symmetry at $\theta=2\pi$ but not at $\theta=0$ (see Ref. [37] and subsequent work in Ref. [33]).

---

## Editorial Decision

resubmitted